# In vitro fusion of single synaptic and dense core vesicles reproduces key physiological properties

Alex J.B. Kreutzberger [iD] [1,2], Volker Kiessling [iD] [1,2], Christopher Stroupe [iD] [1,2], Binyong Liang[1,2], Julia Preobraschenski[3], Marcelo Ganzella[3], Mark A.B. Kreutzberger[4], Robert Nakamoto[1,2], Reinhard Jahn[3], J. David Castle[1,5] & Lukas K. Tamm[1,2]

Regulated exocytosis of synaptic vesicles is substantially faster than of endocrine dense core vesicles despite similar molecular machineries. The reasons for this difference are unknown and could be due to different regulatory proteins, different spatial arrangements, different vesicle sizes, or other factors. To address these questions, we take a reconstitution approach and compare regulated SNARE-mediated fusion of purified synaptic and dense core chromaffin and insulin vesicles using a single vesicle-supported membrane fusion assay. In all cases, Munc18 and complexin are required to restrict fusion in the absence of calcium. Calcium triggers fusion of all docked vesicles. Munc13 (C1C2MUN domain) is required for synaptic and enhanced insulin vesicle fusion, but not for chromaffin vesicles, correlating inversely with the presence of CAPS protein on purified vesicles. Striking disparities in calcium-triggered fusion rates are observed, increasing with curvature with time constants 0.23 s (synaptic vesicles), 3.3 s (chromaffin vesicles), and 9.1 s (insulin vesicles) and correlating with rate differences in cells.

[1] Center for Membrane and Cell Physiology, University of Virginia, Charlottesville, VA, USA. [2] Department of Molecular Physiology and Biological Physics, University of Virginia, Charlottesville, VA, USA. [3] Department of Neurobiology, Max Planck Institute for Biophysical Chemistry, Göttingen, Germany. [4] Department of Biochemistry and Molecular Genetics, University of Virginia, Charlottesville, VA, USA. [5] Department of Cell Biology, University of Virginia, Charlottesville, VA, USA. Correspondence and requests for materials should be addressed to L.K.T. (email: lkt2e@virginia.edu)

I t is well established that calcium-triggered exocytosis of synaptic vesicles is significantly faster than that of large dense-core vesicles in neurons and neuroendocrine cells despite identical SNARE proteins and similar regulatory proteins. These differences are thought to be caused by variations in spatial organization (e.g., with respect to calcium channels), vesicle size, or the precise molecular composition of the release sites, but how exactly these factors determine release kinetics is not known. It is also well known that the final exocytotic step of membrane fusion is catalyzed by specific SNARE proteins including syntaxin-1a and SNAP-25 in the plasma membrane and synaptobrevin-2/VAMP-2 in the secretory vesicle membrane, and that these essential SNARE proteins form the core fusion machinery in neuronal synaptic vesicles, in adrenal chromaffin dense-core vesicles, and in endocrine pancreatic insulin vesicles (also referred to as insulin granules). Moreover, the regulatory proteins Munc18, complexin, synaptotagmin, and a MUN domain-containing protein, such as Munc13 or CAPS (calcium activated protein for secretion) are required in all three cell types to regulate assembly, prevent fusion in the absence of calcium, and prime the SNAREs for rapid fusion in response to elevated levels of calcium[1]. Finally, specific isoforms of the secretory vesicle protein synaptotagmin function as the calcium sensors that trigger the fusion reaction in all three systems[2]. The fact that the protein compositions are so similar in these physiologically very distinct cell types raises the question whether the kinetic differences are an intrinsic property of the release machinery or whether they are caused by more complex cellular factors such as the spatio-temporal properties of the calcium trigger. A recently developed reconstitution system that allows studying docking, priming, and fusion kinetics of vesicles from natural sources in a single-vesicle assay[3] was used to mechanistically investigate kinetic differences of different secretory vesicle types. This assay is ideally suited to answer key questions about the source of the different kinetics and details of the regulation of the docking, priming, and fusion processes by key proteins such as Munc18, complexin, Munc13, synaptotagmin, and CAPS.

Complete biochemical reconstitution approaches starting from all purified components have been powerful to dissect many elements of the mechanisms these proteins utilize to drive fusion[4–8]. SNARE-mediated fusion of secretory vesicles purified from endogenous sources has also been reconstituted into fusion assays with artificial membranes, but most of these fusion assays do not distinguish between docking, priming, and fusion[9–14]. Recently, it was shown that recombinant Munc18 and complexin-1 can be added during reconstitution of dense-core vesicles purified from PC12 cells to generate a docked state that does not proceed to fusion until calcium is injected[3,15]. However, whether other secretory vesicles from different cell types can be arrested in this state and triggered for fusion in a similar manner is still unclear.

To build on this capability and to determine mechanistic differences among secretory vesicle types, the secretory vesicle—reconstituted planar membrane fusion approach was extended to synaptic vesicles and insulin granules. Similar to dense-core vesicles from PC12 cells[3], insulin vesicles can be labeled biosynthetically through the expression of fluorescently tagged proinsulin, which is processed and packaged among the vesicle content[16]. Because synaptic vesicles do not have a secretory protein content, a different strategy was used to achieve internal fluorescent labeling. Capitalizing on the presence of the endogenous proton pump[17], ATP-driven internal acidification enabled concentration of the fluorescent weak base acridine orange within synaptic vesicles. For comparison, this same approach was also employed with purified dense-core vesicles and insulin vesicles enabling simultaneous monitoring of fusion with a fluorescently tagged protein and a small fluorescent reporter. All three vesicle types interact with planar bilayers in a SNARE dependent manner. Addition of Munc18 and complexin-1 to the planar-supported membrane fusion assay arrests all three secretory vesicles in a docked non-fusing state. In the presence of a MUN domain-containing protein, calcium readily triggers fusion of the vesicles from this docked/primed state. In the case of synaptic vesicles, this was achieved by addition of a recombinant C1C2MUN region of Munc13. The purified dense-core vesicles did not have the strict requirement for this recombinant Munc13 construct because of the presence of the endogenous MUN domain-containing proteins CAPS-1 and CAPS-2 present on the vesicle. Insulin vesicle samples contained a small fraction of the total cellular CAPS-2, however they showed an additional kinetic enhancement of calcium-stimulated fusion upon addition of the C1C2MUN fragment. Despite using a similar well controlled docking and priming machinery in the target membrane, each vesicle type has distinct fusion kinetics that directly correlates with its curvature and is independent of synaptotagmin isoform and vesicle SNARE density in the vesicle membrane.

## Results

**Secretory vesicle labeling and single-vesicle fusion.** Dense-core vesicles and insulin vesicles were purified from PC12 and INS1 cells, respectively, using an iso-osmotic fractionation procedure (see Methods, Supplementary Fig. 1b, d, e). The lumen of the dense-core vesicles was fluorescently labeled by expressing neuropeptide-Y tagged to mRuby (NPY-mRuby)[3]. The insulin vesicles were purified from INS1 cells that stably express human proinsulin containing a green fluorescent protein insert in the C-peptide domain (C-peptide-GFP). The expressed proinsulin derivative is transported and processed similarly to endogenous proinsulin, and C-peptide-GFP is stored and secreted in parallel with endogenous insulin providing a fluorescent label of the insulin vesicle content[16]. When purified and examined for fusion with reconstituted planar membranes containing the plasma membrane SNARE proteins by TIRF microscopy, both single dense-core vesicle and single insulin vesicle fusion events can be analyzed by observing similar characteristic fluorescent signals that have been described in detail previously for dense-core vesicles (Fig. 1a, b)[3,18]. In these experiments, asymmetric-supported membranes were prepared containing lipids mimicking the plasma membrane (brain(b)PC:bPE:bPS:chol:PI:bPIP$_2$ at the ratio 25:25:15:30:4:1 on the cytoplasmic mimicking leaflet and 67:30:3 bPC:chol:DMPE-PEG2000-triethoxysilane (DPS) on the extracellular mimicking leaflet) and a minimal target membrane SNARE fusion machinery consisting of syntaxin-1a (183–288 lacking the regulatory H$_{abc}$ domain) and dodecylated SNAP-25 (dSNAP-25)[19]. Upon addition of dense-core vesicles or insulin vesicles to the planar-supported membranes, the peak fluorescence intensity originating from single secretory vesicles increases when the vesicle enters the TIRF field and binds to the planar membrane, decreases transiently at the onset of fusion, then increases rapidly as the vesicle collapses into the planar membrane, and finally decreases toward baseline as the label diffuses away from the site of fusion (Fig. 1a, b, g and Supplementary Fig. 2) as previously demonstrated[3,18].

Synaptic vesicles have been purified from rat brain and characterized in detail[17], and their SNARE-mediated membrane fusion to planar-supported membranes containing the plasma membrane SNAREs has been monitored previously using fluorescent lipids as indicators[10]. The fractionation scheme and western blots of fractions used to obtain purified synaptic vesicles are shown in Supplementary Fig. 1a, c for comparison with the other secretory vesicle purifications. The interior of the synaptic

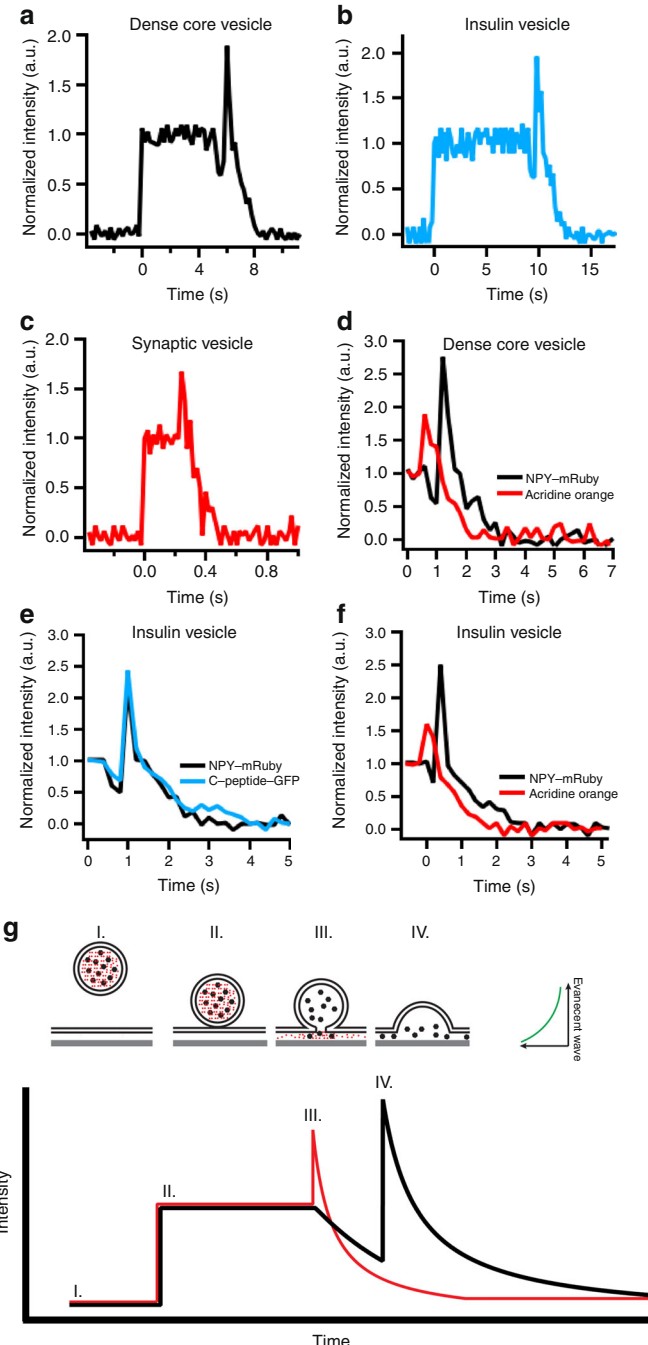

**Fig. 1** Characteristic fusion events of purified secretory vesicles with planar-supported membranes. The intensity traces of the peak intensity pixel of a single secretory vesicle docking and fusing with the planar-supported bilayer containing t-SNAREs are shown for **a** dense-core vesicles labeled with NPY-mRuby, **b** insulin vesicles labeled with C-peptide-GFP, and **c** synaptic vesicles labeled with acridine orange. The single events are representative examples with more example events shown in Supplementary Fig. 2. **d** Intensity traces from a double-labeled (acridine orange, red; NPY-Ruby, black) dense-core vesicle illustrates that during fusion, egress of acridine orange precedes NPY-Ruby by ~200 ms. Intensity traces from insulin vesicles double-labeled with **e** C-peptide-GFP and NPY-mRuby or with **f** NPY-mRuby and acridine orange show that C-peptide-GFP and NPY-mRuby have the same characteristic release behavior while acridine orange starts to be released about ~200 ms before the onset of release of the fluorescent proteins. **g** Schematic drawing of the theoretical intensity traces for single secretory vesicles labeled with a fluorescent protein (black) or acridine orange (red). Intensity traces are of a region of interest over time. I. Initially the labeled secretory vesicle is not within the TIRF field and no fluorescence is observed. II. The vesicle docks to the planar-supported membrane causing an immediate increase of fluorescence intensity in the TIRF field. III. After some time, the vesicle fuses with the planar membrane. The onset of fusion begins with the opening of a small fusion pore resulting in an increase in fluorescence of the small molecule dye acridine orange or a decrease in fluorescence of the fluorescent protein. The decrease in observed fluorescence is due to lateral diffusion of the fluorescent indicator away from the site of fusion. When fusion is observed with a fluorescent protein, the protein is first more slowly released through the initial narrow fusion pore until, at time point IV, the vesicle collapses into the planar membrane as observed by an increase in fluorescence due to the forward movement of the protein within the TIRF field following a continued decrease in intensity as the protein continues to leave the fusion site

the dye amenable for optimal visualization in the planar-supported membrane fusion assay.

A representative fluorescent signal from an acridine orange-loaded synaptic vesicle docking and fusing with a plasma membrane mimicking planar-supported bilayer-containing syntaxin-1a (183–288) and dSNAP-25 is shown in Fig. 1c and more examples are shown in Supplementary Fig. 2a. The abrupt increase in fluorescence at the time indicated as 0 identifies synaptic vesicle docking to the planar-supported membrane. The following increase in fluorescence after ~250 ms indicates the onset of fusion as the acridine orange dye moves close to the evanescent TIRF field and dequenches. The fluorescence intensity first peaks and then decreases as the dye diffuses away from the site of fusion inside the cleft between supported membrane and quartz substrate (Fig. 1c, g and Supplementary Fig. 2a).

Double labeling of dense-core vesicles and insulin vesicles allowed for a direct comparison of release signals originating from different fluorescent reporters. Both dense-core vesicles and insulin vesicles have functional V-ATPase proton pumps, thus can concentrate acridine orange in the presence of ATP and valinomycin. Acidification is reversible by CCCP and inhibited in the presence of bafilomycin (Fig. 2b, c). A fusion event of a dense-core vesicle simultaneously labeled with acridine orange and NPY-mRuby is shown in Fig. 1d. The acridine orange fusion peak begins about 200 ms prior to the initial dip observed with the NPY-mRuby signal. Thus, the initial narrow fusion pore allows the release of small molecules like acridine orange while still blocking the diffusion of the bulkier NPY-mRuby through the pore. After ~200 ms from the onset of the acridine orange peak, the NPY-mRuby signal begins to decrease indicating the pore has widened enough for the protein to diffuse away from the site of

vesicles was loaded with the fluorescent marker acridine orange for fusion studies. This was done by generating a transmembrane proton gradient by means of the V-ATPase proton pump on the synaptic vesicle[17]. In the presence of ATP and the K+-ionophore valinomycin, acridine orange concentrates into the synaptic vesicle interior in response to vesicle lumen acidification, which can be observed by the progressive self-quenching of the acridine orange fluorescence (Fig. 2a)[20]. The addition of the proton pump inhibitor bafilomycin A blocks acridine orange loading, whereas the protonophore CCCP rapidly reverses dye loading (Fig. 2a). Acridine orange loading was optimized for the TIRF planar-supported membrane fusion assay by increasing concentrations of secretory vesicles (relative to the loading experiments) and decreasing the concentration of acridine orange to 0.1 μM making

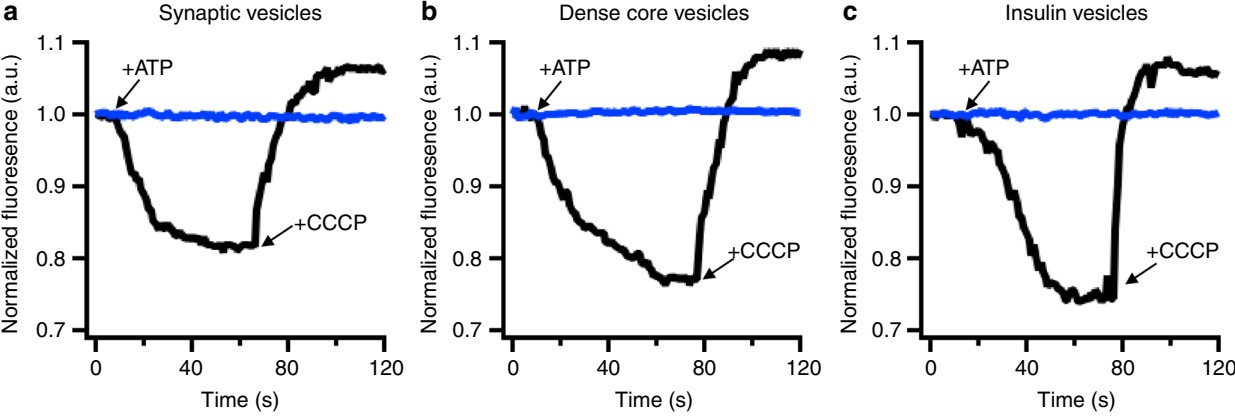

**Fig. 2** Purified secretory vesicles contain functional V-ATPase acidification of **a** synaptic vesicles, **b** dense-core vesicles, or **c** insulin vesicles. Bulk acridine orange fluorescence was monitored in the presence of the secretory vesicle and valinomycin (black line). Addition of ATP drives progressive self-quenching of the acridine orange fluorescence due to concentration into the interior of the secretory vesicle. The addition of CCCP dissipates the pH gradient causing an increase in acridine orange fluorescence. The presence of bafilomycin inhibits the proton pump preventing the acidification of the secretory vesicles (blue traces)

fusion. Approximately 600 ms after the onset of NPY-mRuby diffusion the dense-core vesicle collapses into the planar-supported membrane pulling the remaining NPY-mRuby forward into the TIRF field, which generates the peak fluorescence prior to the final decay of fluorescence due to diffusion away from the site of fusion (see Supplementary Fig. 1 of ref. [3] for further explanation of single-vesicle fusion kinetic line shapes). The overlap of the spectra of GFP and acridine orange prevents their simultaneous imaging. Previously, NPY was shown to concentrate in the same compartment as C-peptide-GFP when expressed in INS1 cells[21]. Purification of insulin granules from cells expressing NPY-mRuby and C-peptide-GFP together showed that these fluorescent markers are released at the same time in the planar-supported membrane fusion assay (Fig. 1e). Labeling insulin vesicles that only expressed NPY-mRuby with acridine orange revealed similar results to those obtained with dense-core vesicles, namely that the acridine orange signal preceded the decrease in NPY-mRuby signal by ~200 ms (Fig. 1f).

**Munc13 dependence of calcium-triggered native vesicle fusion.** The fluorescently tagged proteins in dense-core vesicles and insulin vesicles, and the acridine orange dye in synaptic vesicles were used to quantify docking and fusion in the supported membrane fusion assay. Docking of synaptic vesicles, dense-core vesicles, and insulin vesicles depends on syntaxin-1a and SNAP-25 (Fig. 3a and Supplementary Table 1). Docking is abolished if either or both of these plasma membrane SNAREs are omitted or in the presence of an inhibitor peptide comprising the soluble domain of synaptobrevin-2 (residues 1–96). As compared to truncated syntaxin-1a (residues 183–288), full-length syntaxin-1a (residues 1–288) containing the regulatory $H_{abc}$ domain, decreases the number of docking events, but fusion is still observed. Incubating Munc18 and complexin-1 with the reconstituted plasma membrane SNAREs syntaxin-1a(1-288) and dSNAP-25 allows vesicles to dock while inhibiting fusion in the absence of calcium (Fig. 3a and Supplementary Table 1). The addition of 100 μM calcium to these docked vesicles stimulates fusion with the planar-supported membranes (Fig. 3b, Supplementary Fig. 3, and Supplementary Table 2). The inclusion of a fluorescent dye in the calcium-containing buffer allowed the delay times between calcium arrival and fusion to be measured (Fig. 3b and Supplementary Fig. 4). The measured time from the commencement of calcium buffer arrival to saturation of the buffer

($\Delta t_{Ca}$) was determined for each experiment measured (Supplementary Fig. 4, Supplementary Tables 2 and 3). All conditions had a $\Delta t_{Ca}$ of <0.5 s.

The calcium-stimulated fusion dependence on Munc13 was tested by inclusion of the C1C2MUN domain in the buffer containing the secretory vesicles when they are injected to dock on planar-supported bilayers containing the plasma membrane SNARE complex (syntaxin-1a[1-288] and dSNAP-25), Munc18, and complexin-1 (Fig. 4a–e). As a consequence, the efficiency, and kinetics of calcium-triggered fusion for synaptic vesicles increased markedly upon the addition of C1C2MUN (Fig. 4a, e). Although C1C2MUN addition had very little effect on dense-core vesicle fusion (Fig. 4b, e), there was a significant kinetic effect on insulin vesicle fusion as well as a slightly increased fusion probability (Fig. 4c, e). These effects can be rationalized by the level of CAPS-1/2 in the different purified secretory vesicle preparations. Dense-core vesicle fractions contain readily detectable amounts of both isoforms of CAPS (Supplementary Fig. 1), insulin vesicles contain a small fraction of the total cellular CAPS-2 isoform (Supplementary Fig. 1), whereas the purified synaptic vesicle samples do not contain any CAPS (Supplementary Fig. 1 and [18]). The delay in the kinetics of insulin vesicles indicates an incomplete priming that is overcome by the presence of C1C2MUN changing the kinetics to a first order reaction (Fig. 4c). The sigmoidal shape of the kinetic curve in the absence of added C1C2MUN to insulin vesicles indicates hidden priming steps that may be delayed due to suboptimal amounts of CAPS-2 or the lack of CAPS-1 in our preparations. The requirement of the Munc13 C1C2MUN domain for priming in the absence of CAPS in dense-core vesicles is consistent with a previous observation that the efficiencies and rates of calcium-stimulated fusion of dense-core vesicles purified from CAPS-1/2 knockdown cell lines decreased dramatically, but could be restored by the addition of the C1C2MUN domain of Munc13 (Fig. 4d) (see also ref. [3]).

**Fusion kinetics of secretory vesicles correlate with size.** Comparing the fusion kinetics from the time of calcium arrival for the three secretory vesicle types, in the presence of Munc18, complexin-1, and C1C2MUN revealed striking differences in the rate of fusion. The cumulative distributions of measured delay times were well described by a fit with a single rate constant for each type of vesicle (Fig. 5a). Synaptic vesicle fusion occurs rapidly

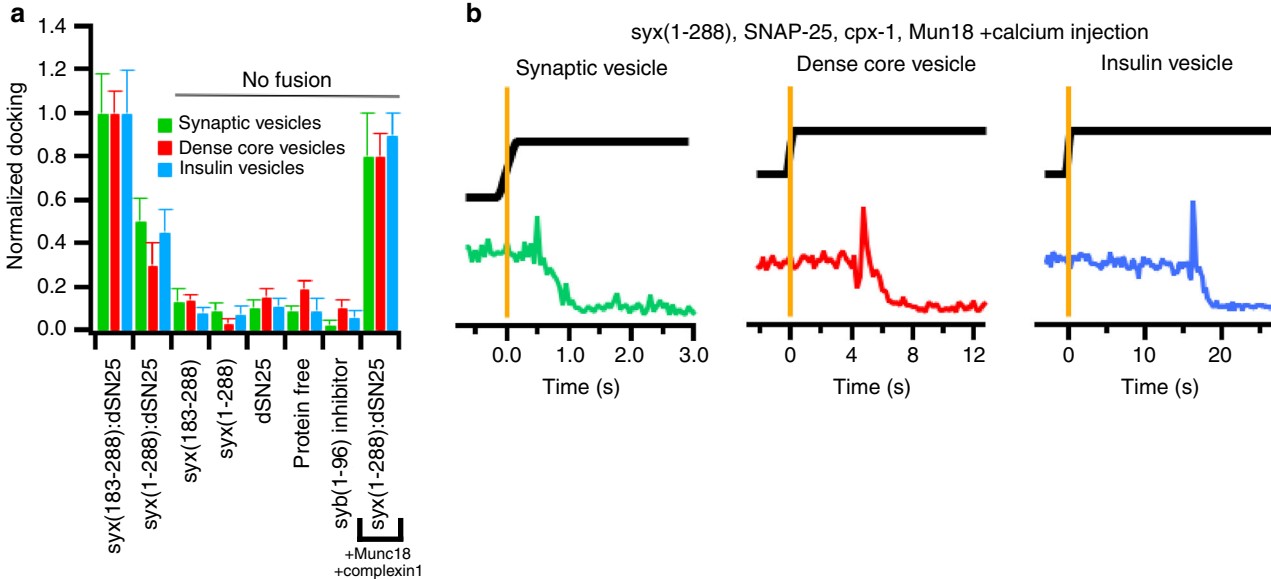

**Fig. 3** Secretory vesicle docking and calcium-triggered fusion. **a** Docking of synaptic vesicles (green), dense-core vesicles (red), and insulin vesicles (blue depends on the presence of both target SNARE proteins syntaxin-1a and SNAP-25 in the planar-supported membrane in the absence of calcium (100 μM EDTA)). Syntaxin-1a (183–288) denotes a construct containing the transmembrane domain and the SNARE motif, whereas syntaxin-1a(1–288) is the full-length protein that also contains the regulatory $H_{abc}$ domain. Secretory vesicle docking to planar-supported lipid bilayers containing different combinations of syntaxin-1a1–288, dSNAP-25, complexin-1, and Munc18. A summary of the statistics is shown in Supplementary Table 1. **b** Single-vesicle fusion events triggered by calcium from a docked state in the presence of syntaxin-1a1–288, dSNAP-25, complexin-1, Munc18, and the C1C2MUN domain of Munc13. Single events for synaptic vesicles, dense-core vesicles, and insulin vesicles are shown. Soluble Alexa647 was included in the buffer containing 100 μM calcium to measure calcium arrival (black trace) to determine delay times of fusion. Orange line is the precise time determined for calcium arrival. The traces represent the release of respective fluorescent content markers of acridine orange for synaptic vesicles (green), NPY-mRuby for dense-core vesicles (red), and C-peptide-GFP (blue) for insulin vesicles. Graphs with expanded times scales around the arrival time of the calcium buffer are shown in Supplementary Fig. 4 and more representative traces are shown in Supplementary Fig. 3. Error bars represent standard errors of the mean

with a characteristic time constant of 0.23 s, whereas dense-core vesicle and insulin vesicle fusion proceeded more slowly with time constants of 3.3 and 9.1 s, respectively, from the time of calcium arrival (Fig. 5a). Determining the kinetics for the fusion reaction after calcium arrival for insulin vesicles labeled with C-peptide-GFP, NPY-mRuby, or acridine orange revealed that the observed delay times are independent of the labeling strategy (Supplementary Fig. 5, Supplementary Table 3).

All three secretory vesicles were reconstituted into the same lipid- and protein-environment in the planar-supported target membranes, indicating that the observed differences in fusion kinetics are caused by the secretory vesicle. The different secretory vesicles contain different isoforms of the calcium sensor for exocytosis, synaptotagmin, namely synaptotagmin-1, -7, and -9. Single isoforms of synaptotagmin were expressed into a previously characterized PC12 cell line lacking endogenous secretory vesicle synaptotagmins. Purifying the secretory vesicles and determining the fusion kinetics from the time of calcium arrival revealed that synaptotagmin-7-containing dense-core vesicles have a slightly faster rate constant than synaptotagmin-1 and synaptotagmin-9 dense-core vesicles (Fig. 5b, Supplementary Fig. 6, and Supplementary Table 4), in agreement with similar data obtained in intact cells[22]. Therefore, the differences in rates caused by different synaptotagmin isoforms cannot explain the observed drastic rate differences between synaptic vesicles, dense-core vesicles, and insulin vesicles.

Another possible explanation for the kinetic difference between the secretory vesicle types might be variations in density of the vesicle SNARE, VAMP-2, on the respective vesicles. It is well known that synaptic vesicles have a high density of VAMP-2, i.e., ~70 copies per vesicle[18], but little is known about the density on

other secretory vesicle types. Previously, the VAMP-2 density was shown to have no effect on fusion rates in a reconstituted proteoliposome fusion assay unless the protein density was below the threshold number required for fusion, i.e., below ~3 copies per 40–50 nm vesicle[23]. To address this concern with respect to the different secretory vesicle types, a soluble plasma membrane Q-SNARE complex (1:1 syntaxin-1a residues 191–253 and SNAP-25, ref. [24]) was used as a competitive inhibitor to block endogenous secretory vesicle VAMP proteins. Addition of this soluble Q-SNARE complex decreased the density of active VAMP on the vesicles and reduced docking of all three types of secretory vesicles to the supported membranes in a dose-dependent manner (Fig. 5c and Supplementary Table 5). When the three types of secretory vesicles were docked in the presence of 0.5 μM soluble Q-SNARE complex, i.e., a concentration that reduced docking by roughly 50%, all three secretory vesicles types were still competent to fuse with the planar membranes when calcium was injected and no effect on the rate of fusion was observed for any of the three secretory vesicle types (Fig. 5d, Supplementary Table 6). Therefore, the VAMP density on the respective vesicles at their respective native densities does not limit fusion as expected from the previous reconstituted proteoliposome fusion studies[23].

The calcium-stimulated, SNARE-mediated fusion rate constants range from $\sim 4.4\,s^{-1}$ for synaptic vesicles to $\sim 0.1\,s^{-1}$ for insulin vesicles (Fig. 5a). So far, we have shown that neither the different synaptotagmin isoforms nor VAMP density can explain these different fusion kinetics. Another obvious difference between these different types of secretory vesicles is their respective size and, therefore, their respective membrane curvature. Modeling of the fusion pathway through several

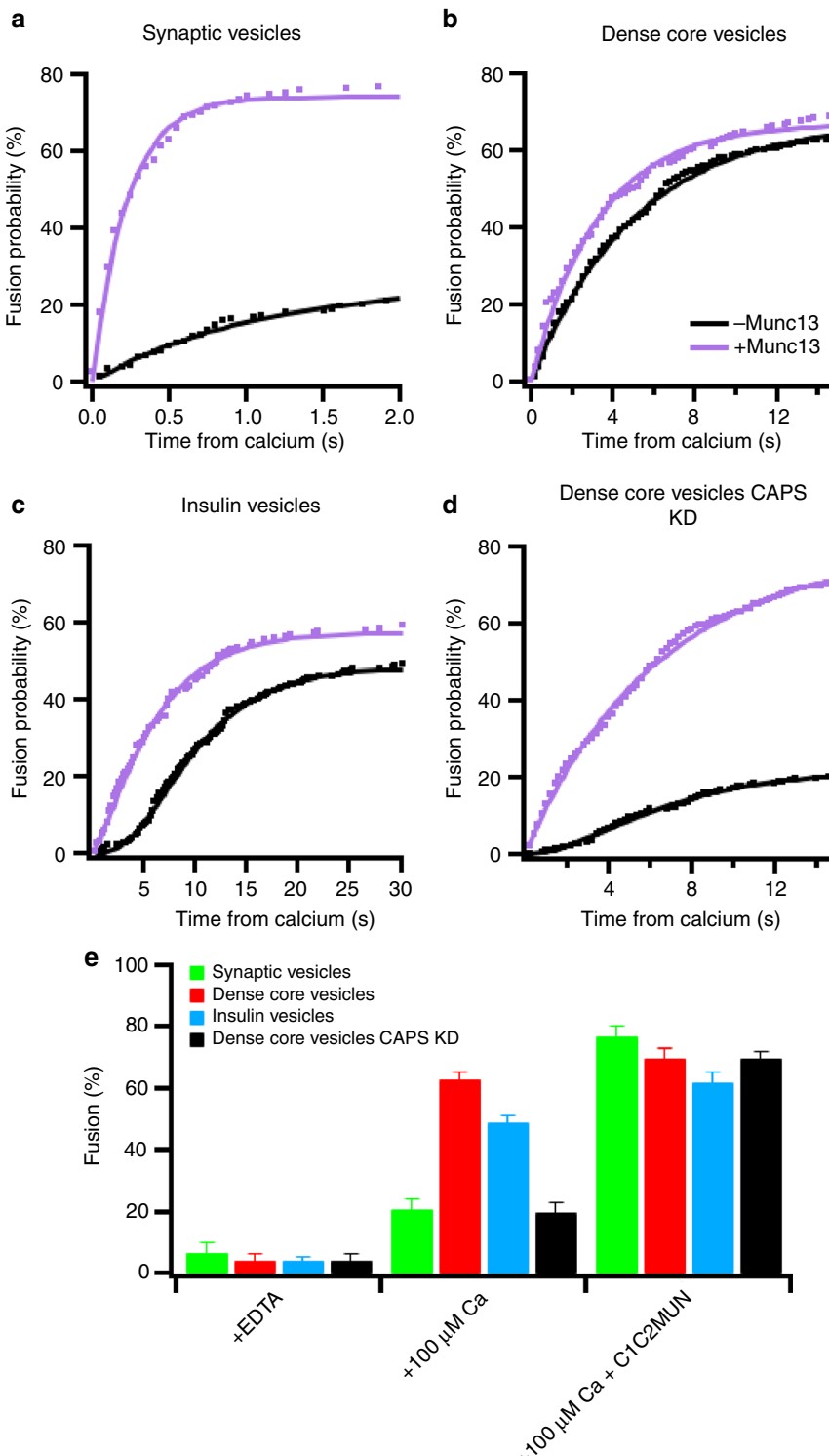

**Fig. 4** Munc13 dependence of secretory vesicle kinetics. Cumulative distribution functions for synaptic vesicles (**a**), dense-core vesicles (**b**), insulin vesicles (**c**), dense-core vesicles lacking endogenous CAPS-1/2 (previously described in ref. [3]) (**d**). All kinetics are triggered from a docked state in the presence of syntaxin-1a:dSNAP-25, Munc18, complexin-1 without (black), or with (purple) 0.2 μM of the C1C2MUN domain fragment of Munc13. Summary statistics for each condition are shown in Supplementary Table 2. **e** Bar graph for the total amount of fusion in the presence of 100 μM EDTA, 100 μM calcium, or 100 μM calcium in the presence of 0.2 μM C1C2MUN domain of Munc13 for each condition. Error bars represent standard errors of the mean

intermediates, Malinin and Lentz[25], revealed that the dependence between the energy barriers of the rate limiting step and the curvature of fusing liposomes is roughly linear. Indeed, when comparing the rate constants of fusion ($k$) with the curvature ($1/R$) of the different secretory vesicles as measured by cryo-electron microscopy (Fig. 5e), this correlation was found to be also valid for the much more complex biological membranes present in purified organelles of different sizes (Fig. 5f).

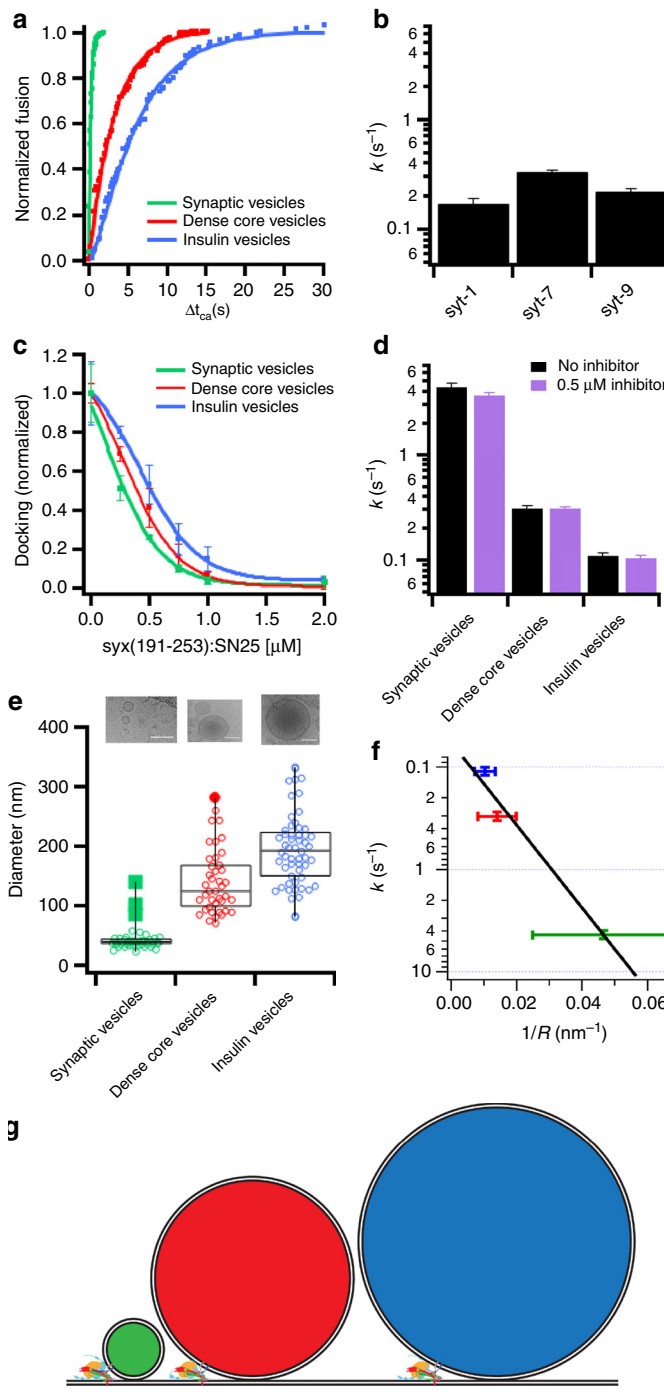

Fig. 5 Secretory vesicle kinetics have a vast range of fusion rates that depend on vesicle size. **a** Cumulative distribution functions of the delay times of fusion from the time of calcium arrival ($\Delta t_{Ca2+}$) for the three secretory vesicle types. Fusion of vesicles is triggered from a docked state in the presence of syntaxin-1a:dSNAP-25, Munc18, complexin-1, and the C1C2MUN domain of Munc13. Summary statistics for each condition are shown in Supplementary Table 2. **b** Rate constants of dense-core vesicles purified from PC12 cell lines depleted of endogenous synaptotagmins and overexpressing single isoforms of synaptotagmin-1, -7, or -9. Vesicles were docked in the presence of syntaxin-1a:dSNAP-25, Munc18, and complexin-1 and triggered to fuse with 100 μM calcium. Cumulative distribution functions for synaptotagmin knock-ins are shown in Supplementary Fig. 6. **c** Docking of all three secretory vesicles is inhibited by a soluble target membrane Q-SNARE complex (syntaxin-1a(191–253):SNAP-25) in the presence of Munc18 and complexin-1 and in the absence of calcium. **d** The rate constant of fusion of all three secretory vesicle types in the absence or presence of 0.5 μM syntaxin-1a(191–253):SNAP-25 is unchanged for calcium-triggered fusion with planar-supported membranes containing syntaxin-1a:dSNAP-25 in the presence of Munc18, complexin-1, and the C1C2MUN domain of Munc13. **e** Vesicle diameter distributions of purified secretory vesicles as measured by cryo-electron microscopy. Example images are shown on top with scale bars being 100 nm. **f** Fitted fusion rate constants ($k$ (s$^{-1}$)) from **a** plotted as a function of the mean curvature (1/$R$), calculated from **e**. The fitted black line assumes that the activation energy depends linearly on the curvature of the fusing vesicles[24]. **g** To-scale model showing the size differences of synaptic vesicles (green), dense-core vesicles (red), and insulin vesicles (blue) docked to a planar target membrane. Error bars represent standard errors of the mean

reconstituted SNARE membranes in the absence of calcium. This behavior is expected based on genetic screens with physiological read-outs in cells, but is often not as robust in fully reconstituted systems as it is in the system employed here. Injection of physiological concentrations of calcium triggers highly efficient fusion of all three types of vesicles from their docked fusion-restricted states given MUN domain-containing co-factors are present.

An important realization concerns the influence of the MUN domain proteins CAPS and Munc13 on calcium-triggered fusion of the three native secretory vesicle types (Fig. 4). Our data on vesicle protein composition (Supplementary Fig. 1) and that of others[17] indicate the variable presence of vesicle associated CAPS —yet, its regulatory role has been broadly debated for dense-core vesicles[30], insulin vesicles[31], and synaptic vesicles[32]. Considering the evidence for at least partial functional overlap of CAPS and Munc13, a clear generic effect of the MUN domain on fusion probability and kinetics has been demonstrated in this work. The characteristic delay observed in the fusion kinetics in the absence of CAPS or Munc13 might indicate their role in catalyzing a fusion competent conformation of a pre-fusion SNARE complex before the arrival of Ca$^{2+}$. Although Ca$^{2+}$ stimulation is still possible without these proteins, it happens with much lower efficiency and with slower kinetics. It is important to note that we previously found that the MUN domain alone only partially rescues the impaired fusion kinetics and that for a full recovery of the reaction the C1C2MUN fragment as well as phosphatidyli-nositol-4,5-biphosphate in the target membrane were required[3]. It is possible that the additional membrane binding domains increase the efficiency of the MUN domain. However, one can also speculate that the Munc13/CAPS-mediated priming of secretory vesicles involves actions on the fusion proteins and the lipid bilayer. The kinetic enhancements that we observe for synaptic vesicles and CAPS-1/2 knockdown dense-core vesicles upon addition of the C1C2MUN fragment of Munc13 has also

## Discussion

The application of the hybrid in vitro assay for calcium-stimulated SNARE-mediated fusion originally developed for dense-core vesicles has been expanded here to synaptic vesicles and insulin granules. The fusion of all three secretory vesicle types has been established to use a similar machinery including the exocytotic SNAREs, syntaxin-1a, and SNARE-25, in conjunction with their regulatory proteins Munc18, complexin-1, Munc13, and/or CAPS[1,2,17,26]. It is important to note that the role of the different SNARE isoforms for insulin exocytosis is debated[27], but several groups have directly implicated syntaxin-1a and SNAP-25 in insulin secretion[26,28,29]. Our data indicate that Munc18 and complexin-1 are sufficient to restrict fusion of all three types of native secretory vesicles with plasma membrane mimicking

been observed in a fully reconstituted model membrane fusion assay[33], but the response of fusion to calcium was less robust than observed here with purified native secretory vesicles.

In this study, the purified recombinant fragment of Munc13 serves as a valuable tool to establish equal degrees of vesicle priming between the three secretory vesicle types. Addition of C1C2MUN allowed us to detect intrinsic differences in fusion between the individual secretory vesicle types. Thus, synaptic vesicles fused rapidly with a time constant of 0.23 s, whereas dense-core vesicles were ~10-fold slower (3.3 s) and insulin vesicles ~30-fold slower (9.1 s). Interestingly, our results correlate with physiological differences between neurons, chromaffin cells, and beta cells where regulated exocytosis of synaptic vesicles and endocrine granules proceeds over quite different timescales, indicating that even the final step in the fusion path preserves this large kinetic range[34–38]. The time constant of 230 ms for synaptic vesicles, although many orders of magnitude faster than time constants measured in bulk reconstituted fusion assays, is still slower than physiological rates of exocytosis in neurons. One contributing factor for this difference is the relatively slow delivery of calcium in our assay. The perfusion delivery speed of the calcium-containing buffer is certainly much slower than the localized rapid rise of calcium concentration via calcium channels near fusion sites in synapses. The calcium perfusion speed in our flow through cell is therefore likely a rate limiting step in our current assay. This influences the rate of synaptic vesicle fusion the most because the $\Delta t_{Ca}$ is on the same time scale as the measured rate constant for fusion (Supplementary Table 2), indicating that synaptic vesicle fusion would likely be still faster in our TIRF-based single-vesicle planar membrane fusion assay system if calcium delivery was instantaneous. Despite this caveat, the relative difference of fusion rates of the three native vesicle systems in an identical experimental format is significant and important.

The underlying cause for the intrinsic rate difference between the three secretory vesicle types is unknown. The differential requirement for the Munc13 construct, due to the presence of different levels of endogenous CAPS between the vesicle types could indicate differences in the protein machineries between the vesicles. Alternatively, other unknown proteins that may be associated with synaptic vesicles may enhance their fusion rate. For example, secretory vesicles of different sources harbor different isoforms of synaptotagmin, such as synaptotagmin-1, -7, and -9 on synaptic vesicles[18], synaptotagmin-1 and -9 on dense-core vesicles, and synaptotagmin-7 and -9 on insulin vesicles. However, measuring the kinetics of fusion from calcium arrival for dense-core vesicles with single knock-ins of each synaptotagmin isoform revealed different synaptotagmin isoforms are not the underlying cause of the kinetic differences of different purified secretory vesicles (Fig. 5b). The vesicle SNARE protein, VAMP-2, is known to be one of the most abundant membrane proteins in synaptic vesicles[17]. Titrations of soluble plasma membrane SNAREs to reduce the effective VAMP density on the secretory vesicles did not affect the rates of fusion from the time of calcium arrival for either type of secretory vesicle (Fig. 5d). This rules out that differences in VAMP density are the driving factor behind the fast rate constant of synaptic vesicles compared to dense-core and insulin vesicles.

Membrane composition is also well documented to affect SNARE-mediated membrane fusion[15,18,23,39–43], and differences in lipid species between the secretory vesicle types could also have a role in observed extents and rates of fusion. One obvious difference between the secretory vesicle types is their size (Fig. 5e). Previously, experimental work has demonstrated membrane curvature to heavily influence SNARE-mediated fusion in fully reconstituted systems[23,43]. Although there are likely many contributing factors for the observed kinetic differences, the theoretical predicted dependence of fusion on membrane curvature[25] seems to be the dominant factor that distinguishes the fusion rates of these three different types of secretory vesicles (Fig. 5f, g).

## Methods

**Materials.** The following materials were purchased and used without further purification: porcine brain L-α-phosphatidylcholine (bPC), porcine brain L-α-phosphatidylethanolamine (bPE), porcine brain L-α-phosphatidylserine (bPS), and L-α-phosphatidylinositol (liver, bovine) (PI), and porcine brain phosphatidylinositol-4,5-bisphosphate (bPIP₂) were from Avanti Polar Lipids; DMPE-PEG2000-triethoxysilan (DPS) was custom-synthesized by Shearwater Polymers (Huntsville, AL); cholesterol, sodium cholate, EDTA, calcium, Opti-Prep Density Gradient Medium, sucrose, MOPS, glutamic acid potassium salt monohydrate, potassium acetate, adenosine 5′-triphosphate (ATP) magnesium (Mg²⁺) salt and glycerol were from Sigma; CHAPS and DPC were from Anatrace; HEPES was from Research Products International; chloroform, ethanol, Contrad detergent, all inorganic acids and bases, and hydrogen peroxide were from Fisher Scientific. Water was purified first with deionizing and organic-free 3 filters (Virginia Water Systems) and then with a NANOpure system from Barnstead to achieve a resistivity of 18.2 MΩ cm⁻¹.

Antibodies for synaptotagmin-1 (mouse monoclonal, catalog number 105 011), synaptotagmin-7 (rabbit polyclonal, catalog number 105 173), and synaptobrevin-2/VAMP-2 (mouse monoclonal, catalog number 104 211) were from Synaptic Systems. The calnexin antibody (rabbit polyclonal, catalog number ADI-SPA-860-D) was from Enzo Life Sciences, the secretogranin II antibody (mouse monoclonal, catalog number 13441-RP01) was from Biodesign International, the GFP antibody (mouse monoclonal, catalog number sc-9996) was from Santa Cruz, and the succinate ubiquinone oxidoreductase antibody (mouse monoclonal) was from Molecular Probes. All antibodies were used at dilutions of 1:1000.

**Dense-core vesicle purification from PC12 cells.** Dense-core vesicle purification was described previously[3]. Pheochromocytoma 12 (PC12) cells (a gift from T. Martin described in ref. [30]) transfected with NPY-mRuby (~20 10-cm plates) were scraped into PBS. Cells were pelleted by centrifugation and then suspended and washed in homogenization medium (0.26 M sucrose, 5 mM MOPS, 0.2 mM EDTA) by pelleting and re-suspending. Following re-suspension in 3 ml of medium containing protease inhibitor (Roche Diagnostics), the cells were cracked open using a ball bearing homogenizer with a 0.2507-inch bore and 0.2496-inch diameter ball. The homogenate was spun at 1500 × g for 10 min at 4 °C in fixed-angle micro-centrifuge to pellet nuclei and larger debris. The post-nuclear supernatant was collected and spun at 8000 × g for 15 min at 4 °C to pellet mitochondria. The post-mitochondrial supernatant was then collected, adjusted to 5 mM EDTA, and incubated 10 min on ice. A working solution of 50% Optiprep (iodixanol) (5 vol. 60% Optiprep: 1 vol. 0.26 M sucrose, 30 mM MOPS, 1 mM EDTA) and homogenization medium were mixed to prepare solutions for discontinuous gradients in Beckman SW55 tubes: 0.5 ml of 30% iodixanol on the bottom and 3.8 ml of 14.5% iodixanol, above which 1.2 ml EDTA-adjusted supernatant was layered. Samples were spun at 190,000 × g for 5 h. A clear white band at the interface between the 30% iodixanol and the 14.5% iodixanol was collected as the dense-core vesicle sample (fraction 9 of western blot shown in Supplementary Fig. 1). The dense-core vesicle sample was then extensively dialyzed in a cassette with 10,000 kDa molecular weight cutoff (24–48 h, 3 × 5 l) into the fusion assay buffer (120 mM potassium glutamate, 20 mM potassium acetate, 20 mM HEPES, pH 7.4). Western blots of the fractionation were performed by taking 0.5 ml fractions from the top of the density gradient. These fractions were pelleted by high speed centrifugation and the pellets were re-suspended in 100 μl of SDS running buffer. SDS-PAGE was run and samples were transferred to nitrocellulose membranes and blotted for secretogranin II (dense-core vesicle marker), synaptotagmin-1 (dense-core vesicle calcium sensor), synaptobrevin-2/VAMP-2 (dense-core vesicle SNARE protein), calnexin (endoplasmic reticulum marker), and succinate ubiquinone (mitochondrial marker).

**Insulin vesicle purification from INS1 cells.** Insulin vesicles were purified from INS1 cells stably expressing human proinsulin-C-peptide-GFP were a gift from P. Arvan and described in ref. [16]. The purification protocol was exactly as described for dense-core vesicles above. Fraction 9 (14.5%/30% iodixanol interface), which was harvested for our experiments, is highly enriched in insulin vesicles that are either mostly or fully mature. Less dense organelles that minimally sediment in 14.5% iodixanol and are prominent in Fraction 3 include condensing vacuoles/earliest immature granules and vesicles derived from ER and Golgi/TGN where proinsulin-GFP is much more enriched relative to C-peptide-GFP, as well as synaptic-like microvesicles and endosomes. Western blots of fractions generated during purification were performed as for dense-core vesicles except that anti-GFP was used to track insulin vesicles. Similar to those of dense-core vesicles, western blots were performed for GFP (insulin vesicle marker), synaptotagmin-7 (insulin vesicle calcium sensor), synaptobrevin-2/VAMP-2 (insulin vesicle SNARE protein),

calnexin (endoplasmic reticulum marker), and succinate ubiquinone oxidor-eductase (mitochondrial marker).

**Synaptic vesicle purification.** Synaptic vesicle purification from rat brain was as previously described[17]. The animals were kept following ethical guidelines approved by the Office of Veterinary Affairs and Consumer Protection of the city of Göttingen Germany (permit number 32.22/Vo). Rat brains were homogenized in buffer (320 mM sucrose, 4 mM HEPES-KOH, pH 7.4) in a glass-Teflon homogenizer. The homogenate was centrifuged at $1000 \times g$ for 10 min. The resulting supernatant was again centrifuged at $15,000 \times g$ for 15 min. Synaptosomes were osmotically lysed with ice-cold water and homogenized with three strokes at 2000 rpm in a homogenizer. The lysate was centrifuged at $25,000 \times g$ for 20 min to obtain LS1 fraction. LS1 fraction was further centrifuged at $200,000 \times g$ for 2 h. The resulting LP2 fraction was suspended in 40 mM sucrose and layered on top of a continuous sucrose gradient (from 0.05 to 0.8 M sucrose) and centrifuged at $65,000 \times g$ for 4 h. Synaptic vesicles were collected at the interface of 0.4–0.6 M sucrose and further purified on a size-exclusion chromatography column (controlled pore glass beads).

**Bulk secretory vesicle acridine orange loading assay.** Secretory vesicles were placed in a cuvette in the presence of 0.5 μM acridine orange and 1 μM valinomycin at room temperature. 1 mM ATP-Mg$^{2+}$ was added to induce quenching of acridine orange due to its concentration in the secretory vesicle interior[20]. Addition of 1 μM CCCP was used to disrupt the proton gradient causing the acridine orange fluorescent signal to increase as the dye leaves the interior of the vesicle. Acidification of secretory vesicles was inhibited by the presence of 0.5 μM bafilomycin.

**Acridine orange loading of vesicles for TIRF microscopy.** Secretory vesicles at high concentration in a volume of 300 μL were incubated in the presence of 0.1 μM acridine orange, 1 μM valinomycin, and 1 mM ATP-Mg$^{2+}$ at room temperature for 10 min. The vesicles were then pelleted in a micro-centrifuge at 12,000 rpm for 5 min. The secretory vesicle pellet was suspended in buffer-containing valinomycin and ATP-Mg$^{2+}$ and then stored on ice. Immediately prior to imaging, vesicles were diluted into normal assay buffer and imaged.

**SNARE protein and complexin purification.** Syntaxin-1a (constructs of residues 183–288 and 1–288), SNAP-25, synaptobrevin-2 (soluble construct residues 1–96), and complexin-1 from *Rattus norvegicus* were expressed in *Escherichia coli* strain BL21(DE3) cells (New England Biolabs, Ipswich, MA) under the control of the T7 promoter in the pET28a expression vector and purified as described previously[19]. Briefly, proteins were purified using Ni-NTA affinity chromatography. After removal of N-terminal His tags by thrombin cleavage, the proteins were purified by ion-exchange or size-exclusion chromatography when necessary. SNAP-25 was quadruply dodecylated through disulfide bonding of dodecyl methanethiosulfonate (Toronto Research Company, Toronto, Ontario) to its four native cysteines[19].

**Munc13 and Munc18 purification.** A protein construct containing the C1, C2B, and MUN homology domains of *R. norvegivus* Munc13–1 (C1C2BMUN, residues: 529-1407, EF, 1453–1531) was expressed in *E. coli* using either the pET28a plasmid originally described in ref. [3] or a variant of it, in which the thrombin cleavage site following an N-terminal his$_6$ tag was replaced with a TEV protease cleavage site. The purification protocols for both constructs were essentially the same except for the cleavage reagents. The new plasmid was made by amplifying the original plasmid with Phusion polymerase (New England Biolabs) and primers with sequence CATCACAGCAGCGGCGAAAACCTGTATTTTCAGGGCATCACCT CGGCCTTG and GCCGCTGCTGTGATGATGATGATGATGGCTGCTGCC, followed by digestion with *Dpn*I and transformation into NEB5alpha cells, using reaction conditions recommended by the manufacturer. Replacement of the thrombin cleavage site with a TEV protease cleavage site was confirmed by Sanger nucleotide sequencing (Genewiz, South Plainfield, NJ). To produce C1C2BMUN, the modified plasmid was transformed into Rosetta2(DE3) cells (Novagen, Burlington, MA). A saturated overnight culture in LB + 50 μg ml$^{-1}$ kanamycin was used to inoculate larger cultures (total of 4 l LB-kanamycin) at 37 °C. When these cultures reached an OD$_{600}$ of ~0.9, the temperature in the incubator was reduced to 18 °C, IPTG (isopropyl-β-D-thiogalactopyranoside) was added to a final concentration of 1 mM, and growth was continued overnight. Cells were harvested, resuspended in 50 ml of NaHEPES pH 7.4, 500 mM NaCl, 20 mM imidazole-HCl pH 8, 10% (v/v) glycerol, 1 mM PMSF, 460 ng ml$^{-1}$ leupeptin, 3.5 μg ml$^{-1}$ pepstatin, 24 μg ml$^{-1}$ AEBSF, frozen dropwise in liquid N$_2$, and stored at −80 °C. Cells were later thawed and lysed by four passes at 18,000 psi in a Microfluidizer M-110L (Microfluidics). Lysate was clarified by centrifugation (40 min at $138,000 \times g$ (42,000 rpm) in a Beckman 45 Ti rotor at 4 °C) and incubated with 10 ml Ni-NTA resin (Qiagen) for 2 h at 4 °C with nutation. The resin was washed with 250 ml of 20 mM NaHEPES pH 7.4, 500 mM NaCl, 20 mM imidazole pH 8, and bound protein was eluted with 20 mM NaHEPES pH 7.4, 100 mM NaCl, 300 mM imidazole-HCl pH 8. EDTA (0.1 mM final), DTT (1 mM final), and TEV protease (1:10 molar ratio) were added and the cleavage reaction was dialyzed against 2 l of 20 mM NaHEPES pH 7.4, 100 mM NaCl, 0.1 mM EDTA, 1 mM DTT at 4 °C overnight, with one buffer change. Cleavage was confirmed by SDS-PAGE and

Coomassie Brilliant Blue staining. Cleaved C1C2BMUN was subjected to anion exchange chromatography (MonoQ 10/100 GL; GE Healthcare), then size-exclusion chromatography (HiLoad 16/60 Superdex 200 prep grade; GE Healthcare) in 20 mM NaHEPES pH 7.4, 150 mM KCl, 1 mM DTT, 10% glycerol. Aliquots were frozen in liquid N$_2$ and stored at −80 °C.

Munc18 (*R. norvegicus* munc18-1) was made essentially as described by Hu et al.[44], except that no detergent was used. Importantly, Munc18 was purified in the presence of 10% glycerol in all buffers and in the final purification step by ion-exchange chromatography (MonoQ). The inclusion of 10% glycerol improved the stability and purity of the protein.

**Planar-supported bilayers with plasma membrane SNAREs.** Planar-supported bilayers with reconstituted plasma membrane SNAREs were prepared by the Langmuir–Blodgett/vesicle fusion technique as described in previous studies[45–47]. Quartz slides were cleaned by dipping in 3:1 sulfuric acid:hydrogen peroxide for 15 min using a Teflon holder. Slides were then rinsed in milli-Q water. The first leaflet of the bilayer was prepared by Langmuir–Blodgett transfer onto the quartz slide using a Nima 611 Langmuir–Blodgett trough (Nima, Conventry, UK) by applying the lipid mixture of 70:30:3 bPC:Chol:DPS from a chloroform solution. Solvent was allowed to evaporate for 10 min, the monolayer was compressed at a rate of 10 cm$^2$ min$^{-1}$ to reach a surface pressure of 32 mN m$^{-1}$. After equilibration for 5 min, a clean quartz slide was rapidly (200 mm min$^{-1}$) dipped into the trough and slowly (5 mm min$^{-1}$) withdrawn, whereas a computer maintained a constant surface pressure and monitored the transfer of lipids with head groups down onto the hydrophilic substrate.

Plasma membrane SNARE-containing proteoliposomes with a lipid composition of bPC:bPE:bPS:Chol:PI:PI(4,5)P$_2$ (25:25:15:30:4:1) were prepared by mixing the lipids and evaporating the organic solvents under a stream of N$_2$ gas followed by vacuum desiccation for at least 1 h. The dried lipid films were dissolved in 25 mM sodium cholate in a buffer (20 mM HEPES, 150 mM KCl, pH 7.4) followed by the addition of an appropriate volume of synatxin-1a and SNAP-25 in their respective detergents to reach a final lipid/protein ratio of 3000 for each protein. After 1 h of equilibration at room temperature, the mixture was diluted below the critical micellar concentration by the addition of more buffer to the desired final volume. The sample was then dialyzed overnight against 1 l of buffer, with one buffer change after ~4 h with Biobeads included in the dialysis buffer. To complete formation of the SNARE-containing supported bilayers, proteoliposomes were incubated with the Langmuir–Blodgett monolayer with the proteoliposome lipids forming the outer leaflet of the planar-supported membrane and most SNAREs oriented with their cytoplasmic domains away from the substrate and facing the bulk aqueous region[45–48]. A concentration of ~77 μM total lipid in 1.2 ml total volume was used. Proteoliposomes were incubated for 1 h and excess proteoliposomes were removed by perfusion with 5 ml of buffer (120 mM potassium glutamate, 20 mM potassium acetate (20 mM potassium sulfate was used in buffers with acridine orange labeled vesicles), 20 mM HEPES, 100 μM EDTA, and pH 7.4).

**Total internal reflection fluorescence microscopy.** Experiments examining single-vesicle docking and fusion events were performed on a Zeiss Axiovert 35 fluorescence microscope (Carl Zeiss), with a ×63 water immersion objective (Zeiss, numerical aperture, 0.95) and a prism-based TIRF illumination. The light source was an OBIS 532 LS laser or an OBIS 488 LS laser from Coherent Inc. Fluorescence was observed through a 610 nm band pass filter (D610/60, Chroma) by an electron multiplying charge coupled device (CCD) (DU-860E, Andor Technology). The electron multiplying CCD (EMCCD) was cooled to −70 °C, and the gain was set at 200. The prism-quartz interface was lubricated with glycerol to allow easy translocation of the sample cell on the microscope stage. The beam was totally internally reflected at an angle of 72° from the surface normal, resulting in an evanescent wave that decays exponentially with a characteristic penetration depth of ~100 nm. An elliptical area of 250 μm × 65 μm was illuminated. The laser intensity, shutter, and camera were controlled by a homemade program written in LabVIEW (National Instruments).

Experiments triggering secretory vesicle fusion with calcium were performed on a Zeiss Axiovert 200 fluorescence microscope (Carl Zeiss) with objective and TIRF setups as described above. The light source was a 488 or 514 nm beamline of an argon ion laser (Innova 90C, Coherent), controlled through an acousto-optic modulator (Isomet), and a diode laser (Cube 640, Coherent) emitting light at 640 nm. The characteristic penetration depths were between 90 and 130 nm. An OptoSplit (Andor Technology) was used to separate two fluorescence spectral bands. Fluorescence signals were recorded by an EMCCD (iXon DV887ESC-BV, Andor Technology). The EMCCD camera was cooled to −70 °C, and the electron gain was set at 200.

**Single-vesicle planar-supported membrane assay.** Single-vesicle planar-supported membrane docking and fusion has been previously described[3,45]. Plasma membrane SNARE proteins-containing planar-supported bilayers were washed with buffer (120 mM potassium glutamate, 20 mM potassium sulfate, 20 mM HEPES, pH 7.4) containing 100 μM EDTA. Secretory vesicles were diluted into 2 ml of buffer and injected into the microscope flow chamber containing the

planar-supported membrane. The fluorescence of the secretory vesicles was recorded with a 532 nm laser for NPY-mRuby and C-peptide-GFP, and a 488 nm laser for acridine orange. After injection, the microscope was focused within 30s before images were taken at 200 ms exposure times for dense-core and insulin vesicles, and 40 ms exposure times for synaptic vesicles. All experiments were performed at room temperature.

Single-vesicle fusion data were analyzed using a homemade program written in LabVIEW (National Instruments). Stacks of images were filtered by a moving average filter. The maximum intensity for each pixel over the whole stack was projected on a single image. Vesicles were located in this image by a single-particle detection algorithm described previously[49]. The central pixel and mean fluorescence intensities of a 5-pixel by 5-pixel area around each identified center of mass were plotted as a function of time for all particles in the image series. The exact time points of docking and fusion were determined from the central pixel similar to previous work[45]. Numbers of docking and fusion events were counted for each membrane. The fusion probability was determined by the number of vesicles that underwent fusion compared with the total number of observed vesicles. Docking was normalized for each secretory vesicle preparation to the number of vesicles that docked in the condition of a bilayer-containing syntaxin-1a (183–288) and dSNAP-25 in the presence of 100 μM EDTA.

**Calcium-triggered single-vesicle fusion assay.** Planar-supported bilayers-containing syntaxin-1a:1-288dSNAP-25 (bulk phase-facing leaflet lipid composition of 25:25:15:30:4:1 bPC:bPE:bPS:Chol:PI:bPIP$_2$) were incubated with 0.5 μM Munc18 and 2 μM complexin-1. Secretory vesicles were then injected while keeping the concentrations of Munc18 and complexin-1 constant, with or without the presence of 0.2 μM C1C2MUN from Munc13 as indicated in the text. Secretory vesicle docking was allowed to occur for ~20 min before the chamber was placed on the TIRF microscope and the microscope was focused on the planar-supported membrane. Fluorescence from the sample was recorded while buffer containing 100 μM calcium was injected with a soluble Alexa647 dye in the buffer to monitor the arrival of calcium at the observation site.

**Cryo-electron microscopy of secretory vesicles.** For each sample, 3.5 μl of purified secretory vesicles were applied to C-flat holey carbon grids (Electron Microscopy Sciences). Using an FEI Vitrobot Mark IV the samples were blotted for 4 s and plunge-frozen into liquid ethane. Images were recorded at a magnification of ×29,000 under low electron-dose conditions (~20 $e^-$ Å$^{-2}$) using a 4 k × 4 k CCD camera (Gatan, Pleasanton, CA) fitted to a Tecnai F20 electron microscope.

## Data availability

Data supporting the findings of this manuscript are available from the corresponding author upon reasonable request. A reporting summary for this article is available as a Supplementary Information file. The source data underlying Figs. 3a, 4, 5a–d and Supplementary Fig. 5 are provided as a Source Data file.

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

## Acknowledgements

We thank Jake Wolpe for helpful discussions. This work was supported by NIH P01 GM-072694 (to L.K.T. and R.J.).

## Author contributions

A.J.B.K. purified proteins, dense-core vesicles, insulin vesicles, and performed experiments. B.L. and C.S. developed protein purification protocols and B.L., C.S. and A.J.B.K. purified proteins. J.P. and M.G. purified synaptic vesicles. R.N. assisted with acridine orange labeling. M.A.K. performed electron microscopy. A.J.K.B., V.K. and L.K.T. designed the study. A.J.K. B., V.K., J.D.C. and L.K.T. wrote and all authors edited the manuscript.

## Additional information

**Competing interests:** The authors declare no competing interests.

