## [Peer Review File · Nature Communications]

Reviewers' Comments:

Reviewer #1:

Remarks to the Author:

In a previous study, Kreutzberger and colleagues (Kreutzberger et al., 2017, Science Advances) showed how Munc18, complexin, synaptotagmins, CAPS, Munc18, and phosphoinositides control dense core vesicle docking and fusion employing an elegant hybrid in vitro assay, containing dense core granules and reconstituted planar lipid bilayers. In the present work they also include native synaptic vesicles and insulin-containing granules and compare their behavior at the single vesicle level. Although interesting, the effects observed upon Munc18, complexin and the C1C2MUN addition are expected and consistent with previous data. The major finding is that the 3 different vesicle populations fuse with different time constants (synaptic vesicles: 0.23 seconds, dense core vesicles: 3.3 seconds, insulin vesicles: 9.1 seconds), despite the presence of the same "plasma membrane" machinery (reconstituted syntaxin1/SNAP25). The cause(s) for the distinct time constants remains unclear, but may reflect differences in vesicle diameter/curvature or in the vesicle priming and fusion machinery, as mentioned by the authors. Novel mechanistic insights are rather limited. Addressing the following points could improve the manuscript.

- Further support for the different time constants would be beneficial. The mobility of the distinct fluorophores within the vesicle lumen, e.g. binding of the fluorescently labelled proteins to a luminal matrix, protein oligomerization, their position in the evanescent field, signal sensitivity/fluorescence dequenching, fusion pore permeability may affect the signals in the TIRF-based assay. For example, Schiffer et al reported in 2017 (Chemical Biology) that a C-peptide mCherry fusion protein is not very efficiently released from MIN6 cells. Thus, loading the 3 different vesicle populations (synaptic vesicles/rat brain, dense core vesicles/PC 12 cells and the insulin vesicles/INS1 cells) with the same fluorophore - acridine orange - would allow a more direct determination of the time constant between the calcium trigger and onset of fusion pore opening. Further, in combination with the fluorescently labeled proteins it could reveal information about the subsequent fusion pore dilation / vesicle collapse. In Figure 1D, the authors already convincingly show the feasibility of this approach for the dense core vesicles, also demonstrating that fusion pore opening and expansion occurs within a short time frame. Presently, such data are lacking for the full set of regulatory proteins and the insulin vesicles.
- Figure S1 shows that the isolated insulin vesicles represent a rather minor fraction of the syt-7/9 and VAMP2-containing vesicle population in the INS cells. This might be due to technical issues (vesicle preparation) or reflect an inherently low content of insulin-containing vesicle in INS1 cells (or the presence of an other VAMP homologue on insulin vesicle?). Thus, to provide further evidence that the isolated vesicles represent mature secretory granules please probe the Western blots with an anti-insulin antibody recognizing endogenous proinsulin and mature insulin. (The processing of the proinsulin-GFP seems to inherently inefficient.)
- Do the distinct vesicle preparations contain endogenous Munc13?
- According to Figure 5B and 5D, it seems that the substitution of endogenous CAPS by C1C2MUN may change the kinetics of dense core vesicle fusion. Please clarify. Are similar data available for the insulin vesicles? It is unclear to which degree CAPS-2 is enriched on insulin vesicles (see Figure S1)

Minor points:

- In Figure 2, please mark the time points when ATP or CCCP were added by arrows.
- In the legend to Figure S3, please specify the meaning of the blue vertical line: Ca²⁺ addition.

Reviewer #2:

Remarks to the Author:

This manuscript by Kreutzberger et al. is a follow-up study of the in vitro single-vesicle fusion assay that the authors have previously reported in Sci. Adv. 3(7):e1603208. (2017). The authors

extended their approach to study of fusion processes of the insulin and synaptic vesicles isolated from INS1 cells and rat brain neurons, respectively. These isolated endogenous vesicles were injected onto a Q-SNARE-reconstituted supported membrane for membrane fusion reactions. The docking events were strictly SNARE protein-dependent (Fig. 3A). In particular, addition of Munc18 and complexin1 to this incubation step clamped the fusion process and inhibited fusion pore opening until a Ca²⁺ influx was introduced (Fig. 3B). By including fluorescent dyes in the Ca²⁺ buffer, the authors measured the latency between the arrival of Ca²⁺ ions and the onset of content release (Fig. 4).

The key conclusion of this report is that despite the same reaction conditions being used, the latency observed for the synaptic vesicles was distributed on a much shorter time scale than those for the dense core and insulin vesicles (Fig. 5). The authors attributed this observation to the different sizes of the isolated vesicles, and proposed that the logarithm of the fusion rate is roughly linear with the curvature (i.e., ~inverse of the vesicle radius) (Fig. 6C).

Given that the experimental method has been essentially demonstrated in the previous 2017 paper, it seems necessary to ask whether the biological or biochemical insights garnered here merit enough for publication in *Science Advances*. Unfortunately, this reviewer feels that the current manuscript largely lacks mechanistic understanding of the observed fusion process (Major comment 1). Also, the key conclusion is not convincingly supported by the experimental data (Major comment 2).

Major comment 1: The authors added various fusion regulators (Munc18, complexin1 and C1C2MUN) to confer the speed and Ca²⁺-sensitivity to the observed *in vitro* fusion process.

1-1. The authors reported a strong clamping effect with Munc18/complexin1. Is it possible to observe a similar clamping effect with either Munc18 or complexin alone? If not, do these two proteins work at the same time? Their binding surfaces on the SNARE proteins largely overlap with one another, and it is difficult to imagine that these two proteins work on a single SNARE complex at the same time (eLife 7:e41771 (2018); Nat. Comm. 9, 2639 (2018)). Or do they work in a sequential order? Can the authors produce a similar level of clamping by adding Munc18 and complexin in a specific order (but not vice versa)?

1-2. The C1C2MUN domains of MUNC13 have been mainly implicated in the opening of Syntaxin Habc domain, thereby helping the docking and priming of vesicles. Interestingly, the authors observed a marked effect of these domains on the last step of the fusion process (i.e., fusion pore opening), which has been previously largely ascribed to the interplay between complexin and synaptotagmin. If the authors provide some insights into the molecular function of the C1C2MUN domain in their Ca²⁺-evoked fusion using their assay, that will be a very important piece of information to the field.

Major comment 2: As written above, a key claim of this report is that the three types of secretory vesicles exhibit a curvature-dependent fusion kinetics, regardless of their protein content. While the results are clear and it is plausible that membrane curvature is the major determinant of the Ca²⁺-triggered fusion kinetics, several other factors cannot be ruled out simply based on the presented results.

This point is briefly touched upon in Discussion, but it could seriously weaken the main conclusion of the manuscript. Specifically, the authors argue (p8 line 162) that the lipid/protein environment of the vesicles and planar membranes were the same for the different types of vesicles. While this is true for the artificial planar membrane and target SNARE proteins, the different vesicles clearly differ in their diverse composition of proteins.

One of the most serious concerns is the surface density of the SNARE proteins. A recent paper,

which studied the fusion pore opening between a single nanodisc and a black membrane (Nature 554, 260 (2018)), showed that the SNARE density is a key biochemical element that governs both the speed and size by which the fusion pores open. Because the synaptic vesicles possess a very high density of VAMP2 (~70 copies of VAMP2 per vesicle), there is an ample possibility that the fast pore opening observed for the synaptic vesicles is largely driven by the high VAMP2 density rather than by the smaller size of the synaptic vesicles. The authors need to assess the surface density of R-SNAREs in each type of isolated vesicles. In particular, it might be possible to increase the density of these R-SNARE proteins in the dense core and insulin vesicles via transient over-expression of the corresponding R-SNAREs in the PC12 and INS1 cells, respectively. The authors may want to examine the latency for fusion pore opening with these modified dense core and insulin vesicles, of which the R-SNARE protein densities are increased to be equivalent (or at least comparable) to that of the synaptic vesicles. This experiment will clarify whether the fusion pore opening is still slower for these larger vesicles, even under the same SNARE densities on both vesicle and planar membranes.

Minor comments:

1. It is difficult to understand how acridine orange dyes are loaded into the synaptic vesicles. Does the ~15% decrease in fluorescence signals shown in Fig. 2 unequivocally indicate dye loading of the synaptic vesicles? The authors may want to provide independent evidences, for example, using NTA (Nanoparticle tracking analysis) that is used for analysis of extracellular vesicles.
2. There are some mismatches in the depicted values between Figure 3 and table 1.
3. In Figure 5, the latency distributions for the dense core and the insulin vesicles show a sigmoidal shape in the absence of C1C2MUN, which disappears with addition of C1C2MUN. What causes such a change in the kinetics?
4. There is a typo in the 176th line: SNARE-25 àSNAP-25.
5. In Fig. 1E inset, description of the plot with a green curve is missing.
6. Fig. S1: Western blot for the CAPS-KD PC12 cells (at least for CAPS proteins) must be provided.

Reviewer #3:

Remarks to the Author:

This manuscript by Kreutzberger et al. takes advantage of a reconstituted exocytosis system they previously developed to measure docking and fusion of three different types of native vesicles: neuronal synaptic vesicles, dense core vesicles containing neuropeptide-Y, and insulin-containing vesicles. They report critical components of each system necessary for docking or fusion. Specifically, they observe changes in calcium dependence and the delay between calcium entry, docking, and fusion. Most significantly, they observe a correlation between the radius of curvature of the three types of vesicles and the delay time from calcium entry to fusion of docked vesicles. The data match a previously proposed model. The findings are important and of broad interest.

Minor concerns:

1. On page 18 the authors report "The fusion probability was determined by the number of vesicles that underwent fusion compared with the total number of observed vesicles." This brings up the issue if there were examples where fusion (spike in fluorescence) were observed but not docking (step increase in fluorescence). If so this could skew their data.
2. In their data tables (S1 and S2) the meaning of the ranges given " \pm " are not listed. Are these SEM or a percent confidence range.
3. For Figure 6, vesicle size was measured by cryo-electron microscopy, although an acceptable technique for the limited size range of their vesicles, dynamic light scattering (DLS) would be the preferable method. If any of the types of vesicles included a second size population that was

larger, cryo-EM may miss it.

Minor corrections or typos

1. Figure 1E is not mentioned in the manuscript text.
2. Figure S1 lists fraction 9 as source for the dense core vesicles from PC12 cells and INS cells. But for the INS cells, it looks like fraction 3 would be more appropriate. Is this a typo, or can the authors address why fraction 9 was selected over fraction 3?
3. Figure S2 has many sections, often with different x-axis scales. Although this is also true for Figures 4-5, it is particularly distracting in trying to compare the data in Figure S2. The way it is done in Figure S3 works well.
4. Julia Preobraschenski is not mentioned under "Author Contributions"

Response to Reviewers

Reviewer #1

In a previous study, Kreutzberger and colleagues (Kreutzberger et al., 2017, Science Advances) showed how Munc18, complexin, synaptotagmins, CAPS, Munc18, and phosphoinositides control dense core vesicle docking and fusion employing an elegant hybrid in vitro assay, containing dense core granules and reconstituted planar lipid bilayers. In the present work they also include native synaptic vesicles and insulin-containing granules and compare their behavior at the single vesicle level. Although interesting, the effects observed upon Munc18, complexin and the C1C2MUN addition are expected and consistent with previous data. The major finding is that the 3 different vesicle populations fuse with different time constants (synaptic vesicles: 0.23 seconds, dense core vesicles: 3.3 seconds, insulin vesicles: 9.1 seconds), despite the presence of the same “plasma membrane” machinery (reconstituted syntaxin1/SNAP25). The cause(s) for the distinct time constants remains unclear, but may reflect differences in vesicle diameter/curvature or in the vesicle priming and fusion machinery, as mentioned by the authors. Novel mechanistic insights are rather limited. Addressing the following points could improve the manuscript.

1. Further support for the different time constants would be beneficial. The mobility of the distinct fluorophores within the vesicle lumen, e.g. binding of the fluorescently labelled proteins to a luminal matrix, protein oligomerization, their position in the evanescent field, signal sensitivity/fluorescence dequenching, fusion pore permeability may affect the signals in the TIRF-based assay. For example, Schiffer et al reported in 2017 (Chemical Biology) that a C-peptide mCherry fusion protein is not very efficiently released from MIN6 cells. Thus, loading the 3 different vesicle populations (synaptic vesicles/rat brain, dense core vesicles/PC 12 cells and the insulin vesicles/INS1 cells) with the same fluorophore - acridine orange - would allow a more direct determination of the time constant between the calcium trigger and onset of fusion pore opening. Further, in combination with the fluorescently labeled proteins it could reveal information about the subsequent fusion pore dilation / vesicle collapse. In Figure 1D, the authors already convincingly show the feasibility of this approach for the dense core vesicles, also demonstrating that fusion pore opening and expansion occurs within a short time frame. Presently, such data are lacking for the full set of regulatory proteins and the insulin vesicles.

We appreciate these very constructive comments. In response, we performed several new experiments to address the concerns that the C-peptide-GFP may give a different fusion read out than other fluorescent content markers in the planar supported membrane assay.

1. We double-labeled insulin vesicles with C-peptide-GFP and NPY-mRuby and imaged fusion simultaneously to confirm that there was no difference between these two markers to read out fusion (new Figure 1E & F). We then transfected NPY-mRuby into INS1 cells lacking the C-peptide-GFP and labeled the purified granules with acridine orange to confirm that the differences between the small molecule marker acridine orange and a fluorescent protein marker are the same in insulin vesicles as shown before in dense core vesicles.

2. We determined the kinetics of insulin vesicle fusion from the time of calcium arrival for each of the three labeling strategies separately and obtained the same rate constants for acridine orange, NPY-mRuby, and C-peptide-GFP in insulin granules, new Figure S5 and new Table S3.

2. Figure S1 shows that the isolated insulin vesicles represent a rather minor fraction of the syt-7/9 and VAMP2-containing vesicle population in the INS cells. This might be due to technical issues (vesicle preparation) or reflect an inherently low content of insulin-containing vesicle in INS1 cells (or the presence of an other VAMP homologue on insulin vesicle?). Thus, to provide further evidence that the isolated vesicles represent mature secretory granules please probe the Western blots with an anti-insulin antibody recognizing endogenous proinsulin and mature insulin. (The processing of the proinsulin-GFP seems to inherently inefficient.)

The reviewer is correct that, as compared to primary pancreatic beta cells, there is inherently a low content of insulin vesicles in INS1 cells, including the derived cell line expressing proinsulin with GFP inserted in the C-peptide. Thus, a relatively small fraction of total syts 7/9 and VAMP (~20-25%) in our cells are granule-associated. The remainder is distributed among endosomes, Golgi, endoplasmic reticulum, and synaptic-like microvesicles, which are not resolved in the lower-density fractions of the Optiprep purification.

Regarding the possibility of inefficient processing of proinsulin that is GFP-labeled within the C-peptide, previous studies have shown that the kinetics of processing and transport are very similar to those of endogenous proinsulin (e.g., see Fig 4 in Hussain et al, 2018 MBC 29: 1238-57 and Fig 1 in Haataja et al 2013 JBC 288: 1896-1906). Importantly, in fraction 9 of our Optiprep gradient used for in vitro fusion studies, it is very unlikely that there are GFP sources other than insulin granules for the following reasons. First, as can be seen from the Western blots in Fig S1, we do not detect contamination by endoplasmic reticulum (marked by calnexin), a potential major source of GFP-proinsulin. Second, a small amount of GFP-labeled C-peptide is seen in the low-density fractions at the top of the gradient. This corresponds to low-density early immature granules, which are well established to be the sites where proinsulin begins to be proteolytically processed to insulin and C-peptide (Orci, Cell 1985). To our knowledge there are no other organelles where GFP can accumulate, and the granules present in fraction 9 include less mature as well as fully mature granules. As proinsulin processing to insulin and C-peptide continues during the maturation process, it is not surprising that one can see bands in Western blots that correspond to GFP-labeled proinsulin and GFP-labeled C-peptide.

We also note that in the cells used, GFP colocalizes very well with both expressed NPY-mCherry and endogenous insulin (detected by immunostaining) as had been shown by Hussain et al, 2018 MBC 29: 1238-57 in their Fig S3, thus addressing the co-localization concern of the reviewer.

Finally, the GFP secretory events that we observe and characterize are essentially identical to events characterized by tracking NPY-Ruby expressed in the same granules (new Fig 1E). For all these reasons, we do not feel that adding Western blots for endogenous proinsulin/insulin will increase the insight of our message beyond its present level.

3. Do the distinct vesicle preparations contain endogenous Munc13?

Munc13 does not fractionate with secretory vesicles as has been shown previously by Kreutzberger et al. 2017 *Sci Adv.* 3(7):e1603208 for DCVs and Takamori et al. 2006 *Cell* 127:831-846 for SVs.

4. According to Figure 5B and 5D, it seems that the substitution of endogenous CAPS by C1C2MUN may change the kinetics of dense core vesicle fusion. Please clarify. Are similar data available for the insulin vesicles? It is unclear to which degree CAPS-2 is enriched on insulin vesicles (see Figure S1).

This is indeed an interesting observation. In now renamed Figure 4B we added C1C2MUN on top of endogenous CAPS and it further accelerated the first order kinetics of DCV fusion. The CAPS knock down DCV fusion in renamed Figure 4D loses first order kinetics and becomes sigmoidal and less efficient. Full first order kinetics can be recovered by addition of C1C2MUN. A very reasonable explanation of this behavior is that a function of the endogenous CAPS and the C1C2MUN could be the mediation of specific priming steps that are completed before calcium is added. In the absence of these priming factors, these steps appear to happen after the arrival of calcium causing a kinetic delay expressed in the sigmoidal shape. Even if this is not exactly the correct interpretation, the characteristic delay of the kinetic curve reveals hidden steps in the CAPS KD experiments. Obviously, this is an interpretation of our data, and at present we don't know what the molecular details of the priming reaction(s) are. We are careful in our manuscript to describe our results in this fashion.

The fact that C1C2MUN further enhances the kinetics in the WT vesicles could be stoichiometric effects since we don't know if the co-purified endogenous CAPS on the DCVs reflect the native stoichiometry in the cell and/or if it could reflect a priming reaction that acts on the target membrane where supplementing C1C2MUN may have a more potent effect than CAPS. We are not sure if this same effect of C1C2MUN on insulin vesicle fusion (renamed Figure 4C) is due to a lack of the CAPS-1 isoform or if it is due to the amount of CAPS-2 present. Again, this is speculation and we added discussion about this on p. 9 of the manuscript.

Dense core vesicle fractions contain readily detectable amounts of both isoforms of CAPS (Figure S1), insulin vesicles contain only a small fraction of the total cellular CAPS-2 isoform (Figure S1), while purified synaptic vesicles do not contain any CAPS (Takamori et al. 2006 *Cell* 127:831-846). The kinetic delay of insulin vesicle fusion indicates incomplete priming that is overcome by the presence of C1C2MUN such that the shortened delay times report fusion with apparent first order kinetics (Figure 4C).

Minor points:

1. In Figure 2, please mark the time points when ATP or CCCP were added by arrows. This has been done.
2. In the legend to Figure S3, please specify the meaning of the blue vertical line: Ca²⁺ addition.

The line indicates the time that calcium was determined to have arrived. The figure legend has been modified.

Reviewer #2

This manuscript by Kreutzberger et al. is a follow-up study of the in vitro single-vesicle fusion assay that the authors have previously reported in *Sci. Adv.* 3(7):e1603208. (2017). The authors extended their approach to study of fusion processes of the insulin and synaptic vesicles isolated from INS1 cells and rat brain neurons, respectively. These isolated endogenous vesicles were injected onto a Q-SNARE-reconstituted supported membrane for membrane fusion reactions. The docking events were strictly SNARE protein-dependent (Fig. 3A). In particular, addition of Munc18 and complexin1 to this incubation step clamped the fusion process and inhibited fusion pore opening until a Ca²⁺ influx was introduced (Fig. 3B). By including fluorescent dyes in the Ca²⁺ buffer, the authors measured the latency between the arrival of Ca²⁺ ions and the onset of content release (Fig. 4).

The key conclusion of this report is that despite the same reaction conditions being used, the latency observed for the synaptic vesicles was distributed on a much shorter time scale than those for the dense core and insulin vesicles (Fig. 5). The authors attributed this observation to the different sizes of the isolated vesicles, and proposed that the logarithm of the fusion rate is roughly linear with the curvature (i.e., ~inverse of the vesicle radius) (Fig. 6C).

Given that the experimental method has been essentially demonstrated in the previous 2017 paper, it seems necessary to ask whether the biological or biochemical insights garnered here merit enough for publication in *Science Advances*. Unfortunately, this reviewer feels that the current manuscript largely lacks mechanistic understanding of the observed fusion process (Major comment 1). Also, the key conclusion is not convincingly supported by the experimental data (Major comment 2).

As we have attempted to make clear in the introduction to the paper, we believe our new findings represent significant advances for the field. Yes, the technology is similar to that developed for DCVs in our previous *Science Advances* paper and extended here to SVs and insulin vesicles, but that identical fusion machineries on the target membrane side and supplemented with identical soluble protein factors can explain vastly different fusion rates previously observed in very different cell types is completely novel and important for the field. To our knowledge, this is the first study ever comparing these three different cellular vesicle types with exactly the same methodology. We comment below about major comments 1 and 2.

Major comment 1: The authors added various fusion regulators (Munc18, complexin1 and C1C2MUN) to confer the speed and Ca²⁺-sensitivity to the observed in vitro fusion process.

1-1. The authors reported a strong clamping effect with Munc18/complexin1. Is it possible to

observe a similar clamping effect with either Munc18 or complexin alone? If not, do these two proteins work at the same time? Their binding surfaces on the SNARE proteins largely overlap with one another, and it is difficult to imagine that these two proteins work on a single SNARE complex at the same time (eLife 7:e41771 (2018); Nat. Comm. 9, 2639 (2018)). Or do they work in a sequential order? Can the authors produce a similar level of clamping by adding Munc18 and complex in a specific order (but not vice versa)?

Several aspects of what the reviewer is asking have been characterized in quite some detail in Kreutzberger et al. 2017 Sci Adv. 3(7):e1603208. In a nutshell, Munc18 alone increased vesicle docking and fusion in the presence of syntaxin(1-288):SNAP-25, while complexin alone blocks docking of the vesicles to the supported membranes. The absence of dSNAP-25 in the supported membrane abolishes secretory vesicle docking as shown here (Figure 3) and in the earlier paper. In Figure S7 of the Sci Adv. paper it is shown that, in the presence of Munc18, vesicles dock but do not fuse in the absence of dSNAP25, in agreement with the dual binding of Munc18 to syntaxin and VAMP2 as in the eLife paper that the reviewer referred to (7:e41771), which is a model that had previously been proposed by the Hughson lab (Baker et al. 2015 Science 349:1111). The Nat Comm. paper mentioned by the reviewer explores force clamping of the SNARE four-helix bundle with bound complexin. These experiments were conducted in the absence of a lipid bilayer with a complex that may exist *after* fusion. We showed previously that complexin can bind syntaxin-1a and SNAP-25 alone if the proteins are strictly in a 1:1 acceptor Q-SNARE complex as encountered *before* fusion. The arrangements of the proteins proposed in these references certainly exist under some conditions, but they may not be the starting conditions for fusion that we believe we have defined pretty well in this and several of our previous papers.

The question about the sequence of the assembly reactions between SNARE proteins, Munc18, and complexin is indeed very interesting, but in our opinion has not been explored in sufficient detail in the current literature. It was not the goal of the current study to explore this question in great detail and it will require significant new work beyond the scope of the current paper to address how primed target membrane acceptor complexes are assembled. Interesting initial studies to address this question are provided by Jakhanwal et al 2017 EMBO J 36:1788-1802, Wang et al 2019 Nat Commun 10:69, and Zhou et al 2019 Cell Rep 26:3347-59, but more work needs to be done. The present fusion assay with different vesicle types is perhaps not the best approach to address this interesting question.

1-2. The C1C2MUN domains of MUNC13 have been mainly implicated in the opening of Syntaxin Habc domain, thereby helping the docking and priming of vesicles. Interestingly, the authors observed a marked effect of these domains on the last step of the fusion process (i.e., fusion pore opening), which has been previously largely ascribed to the interplay between complexin and synaptotagmin. If the authors provide some insights into the molecular function of the C1C2MUN domain in their Ca²⁺-evoked fusion using their assay, that will be a very important piece of information to the field.

We agree that information of the molecular function of the C1C2MUN domain would be an important piece of information to the field and we are actively working on this question. However, we have to point out that the effect we see is on the kinetics of the reactions between

Ca²⁺ arrival and pore opening, which is consistent with the implicated role of C1C2MUN in priming. From our data we do not and cannot conclude that C1C2MUN is directly involved in fusion pore opening.

Major comment 2: As written above, a key claim of this report is that the three types of secretory vesicles exhibit a curvature-dependent fusion kinetics, regardless of their protein content. While the results are clear and it is plausible that membrane curvature is the major determinant of the Ca²⁺-triggered fusion kinetics, several other factors cannot be ruled out simply based on the presented results.

This point is briefly touched upon in Discussion, but it could seriously weaken the main conclusion of the manuscript. Specifically, the authors argue (p8 line 162) that the lipid/protein environment of the vesicles and planar membranes were the same for the different types of vesicles. While this is true for the artificial planar membrane and target SNARE proteins, the different vesicles clearly differ in their diverse composition of proteins.

One of the most serious concerns is the surface density of the SNARE proteins. A recent paper, which studied the fusion pore opening between a single nanodisc and a black membrane (Nature 554, 260 (2018)), showed that the SNARE density is a key biochemical element that governs both the speed and size by which the fusion pores open. Because the synaptic vesicles possess a very high density of VAMP2 (~70 copies of VAMP2 per vesicle), there is an ample possibility that the fast pore opening observed for the synaptic vesicles is largely driven by the high VAMP2 density rather than by the smaller size of the synaptic vesicles. The authors need to assess the surface density of R-SNAREs in each type of isolated vesicles. In particular, it might be possible to increase the density of these R-SNARE proteins in the dense core and insulin vesicles via transient over-expression of the corresponding R-SNAREs in the PC12 and INS1 cells, respectively. The authors may want to examine the latency for fusion pore opening with these modified dense core and insulin vesicles, of which the R-SNARE protein densities are increased to be equivalent (or at least comparable) to that of the synaptic vesicles. This experiment will clarify whether the fusion pore opening is still slower for these larger vesicles, even under the same SNARE densities on both vesicle and planar membranes.

This is a valid point. In order to address how SNARE density on the different vesicles could affect our results, we added several new experiments that are shown in our new Figure 5C & D. We pre-incubated each secretory vesicle type with different concentrations of pre-formed binary soluble plasma membrane Q-SNARE complex (syntaxin-1a residues 191-253 and SNAP-25). Injecting vesicles inhibited with this complex greatly reduced the number of observed binding events in a 20 minute time period. At a concentration of 0.5 μ M syntaxin(191-253):SNAP-25, we observed that binding was inhibited by about half for each vesicle type. At this concentration of added Q-SNARE complex, calcium was injected to observe if the vesicles that docked were still fusion competent. We found that neither the fusion efficiency nor the fusion kinetics of Ca²⁺ triggered fusion in our assay depended on the active R-SNARE density on the vesicle for all three types of vesicles, excluding the possibility that R-SNARE density determines the observed fusion rates on the three vesicle types. This observation is also consistent with previously published data where SNARE-mediated liposome fusion was independent of the

SNARE density, as long as the vesicles contained a minimal critical number of SNAREs far below the numbers known to exist on synaptic, dense core, and insulin vesicles (Hernandez et al 2014 PNAS 111, 12037-42).

Minor comments:

1. It is difficult to understand how acridine orange dyes are loaded into the synaptic vesicles. Does the ~15% decrease in fluorescence signals shown in Fig. 2 unequivocally indicate dye loading of the synaptic vesicles? The authors may want to provide independent evidences, for example, using NTA (Nanoparticle tracking analysis) that is used for analysis of extracellular vesicles.

We believe that the ATP dependence and CCCP reversal of loading as well as the observed fusion behavior with acridine orange loaded vesicles is convincing and cannot be easily explained in any other way. The general protocol is not new and has been used by many others in the field to load vesicles with acridine orange.

2. There are some mismatches in the depicted values between Figure 3 and table 1.

Thank you for pointing this out. The figure has been corrected.

3. In Figure 5, the latency distributions for the dense core and the insulin vesicles show a sigmoidal shape in the absence of C1C2MUN, which disappears with addition of C1C2MUN. What causes such a change in the kinetics?

The sigmoidal shape in the now renamed Figure 4 indicates that there are hidden kinetic steps probably due to the absence of a priming protein such as CAPS1/2 or Munc13. Please see more detailed answer to a question by reviewer 1.

4. There is a typo in the 176th line: SNARE-25 àSNAP-25.

This has been corrected.

5. In Fig. 1E inset, description of the plot with a green curve is missing.

This graph is the evanescent wave of the TIRF field. The green curve is now labeled in renamed panel G.

6. Fig. S1: Western blot for the CAPS-KD PC12 cells (at least for CAPS proteins) must be provided.

This cell line was generated and described in Kreutzberger et al. 2017 Sci Adv. and the requested blots are presented in Figure S4 of that paper. The reference has been added to the figure legend.

Reviewer #3

This manuscript by Kreutzberger et al. takes advantage of a reconstituted exocytosis system they previously developed to measure docking and fusion of three different types of native vesicles: neuronal synaptic vesicles, dense core vesicles containing neuropeptide-Y, and insulin-

containing vesicles. They report critical components of each system necessary for docking or fusion. Specifically, they observe changes in calcium dependence and the delay between calcium entry, docking, and fusion. Most significantly, they observe a correlation between the radius of curvature of the three types of vesicles and the delay time from calcium entry to fusion of docked vesicles. The data match a previously proposed model. The findings are important and of broad interest.

Thank you for the positive assessment of our work.

Minor concerns:

1. On page 18 the authors report “The fusion probability was determined by the number of vesicles that underwent fusion compared with the total number of observed vesicles.” This brings up the issue if there were examples where fusion (spike in fluorescence) were observed but not docking (step increase in fluorescence). If so this could skew their data.

Events that occur at the moment of docking are observable because, in this case, one sees a decrease in fluorescence as the dye diffuses away from the docking site. Therefore, there is no concern of this affecting the docking counts in Figure 3A.

2. In their data tables (S1 and S2) the meaning of the ranges given “ \pm ” are not listed. Are these SEM or a percent confidence range.

These are standard errors of the mean. This is now stated in the header of all tables.

3. For Figure 6, vesicle size was measured by cryo-electron microscopy, although an acceptable technique for the limited size range of their vesicles, dynamic light scattering (DLS) would be the preferable method. If any of the types of vesicles included a second size population that was larger, cryo-EM may miss it.

We appreciate the reviewer’s opinion but feel that it is a matter of opinion if DLS is better than cryoEM for estimating vesicle size distributions. DLS weighs the largest fractions most heavily, which may skew the results, whereas very large vesicles may be missed by cryoEM if they are very rare.

Minor corrections or typos:

1. Figure 1E is not mentioned in the manuscript text.

It is now mentioned (renamed Figure 1G).

2. Figure S1 lists fraction 9 as source for the dense core vesicles from PC12 cells and INS cells. But for the INS cells, it looks like fraction 3 would be more appropriate. Is this a typo, or can the authors address why fraction 9 was selected over fraction 3?

Fraction 3 are lighter particles that are not mature insulin vesicles. The top band of GFP is unprocessed pro-insulin-GFP and the lower band is C-peptide-GFP (the top band is stronger in the lower density fractions). Also, there are other lighter membrane fractions, like synaptic-like microvesicles that contain VAMP-2 and synaptotagmins. Mature insulin vesicles have a dense core that causes them to be in the dense fraction 9. Additional text in the materials and methods (on pg. 16 under insulin vesicle purification) was added to further clarify this.

3. Figure S2 has many sections, often with different x-axis scales. Although this is also true for Figures 4-5, it is particularly distracting in trying to compare the data in Figure S2. The way it is done in Figure S3 works well.

This is unfortunately unavoidable. The delay times between when docking and fusion occur are drastically different for the different secretory vesicles. This necessitates different time scales to show events for synaptic vesicles, dense core vesicles, and insulin vesicles. However, the renamed Figure 5A shows the kinetic curves from all vesicle types together in one graph.

4. Julia Preobraschenski is not mentioned under “Author Contributions”
Julia Preobraschenski has been added to this list.

Additional Changes not Requested by Reviewers

To improve clarity and to accommodate the several new figure panels requested by the reviewers and mentioned above, we have mildly reorganized our figures.

Figure 1: panels E and F describe new experiments performed in response to reviewers;
former panel E is now G

Figure 2: panel B is new, in response to reviewers

Figure 3: former panel B data reorganized and subsumed in new panel E of new Figure 4;
former Figure 4 is now new Figure 3B

Figure 4: panels A-D are the same as panels A-D of former Figure 5;
panel E contains data of former Figure 3B enhanced with additional data DCV CAPS knockdown

Figure 5: panels A, E, F, and G are identical to the 4 panels of the previous Figure 6;
Panels B, C, and D contain new data describing experiments performed in response to the reviewers

Tables S3-S6 are new; some describe new data of experiments performed in response to reviewers

Figure S1: Vesicle purification schemes have been added as panels A and B to enhance clarity of our procedures; although not new data panel C has been added for completeness. Panels D and E are former panels A and B.

Figures S2 – S4 are unchanged

Figure S5 is new

Reviewers' Comments:

Reviewer #1:

Remarks to the Author:

In their revised manuscript, Kreutzberger and colleagues have satisfactorily addressed my previous concerns.

Minor point: In Figure 3B and Figure S4 please explain vertical orange and blue lines, respectively.

Reviewer #2:

Remarks to the Author:

The revised manuscript by Kreutzberger et al. addressed most of the concerns previously raised by the reviewers. In particular, the knock-in of single Syt isoforms in Fig. 5B and the titration of available v-SNAREs using soluble t-SNARE inhibitors (in Fig. 5C and D) demonstrate the power of this hybrid assay in dissection of the molecular mechanisms underlying the Ca²⁺-evoked exocytosis. This reviewer has only a few minor comments that the authors may find helpful in final preparation of their manuscript for publication in Nature Communications.

1. With the new experimental data using the soluble t-SNARE inhibitors, the authors suggest that the v-SNARE density may not critically determine the latency before the Ca²⁺-triggered fusion pore opening (Fig. 5C and D). On the other hand, this latency time dramatically depends on the presence of the C1C2MUN domain of Munc13 (Fig. 4). The current understanding in the field is that these MUNC13 domains work to facilitate initial assembly of the neuronal SNARE complexes. If the v-SNARE complex density (and probably, the number of neuronal SNARE complexes) is not a determining factor of the latency, this reviewer wonders what could be the molecular mechanism by which the MUNC13 domains shortens the latency. The authors may want to comment on and discuss this issue.

2. In Discussion, the authors argue that the fusion pore opening, in particular, of the synaptic vesicles was rate-limited by slow delivery of Ca²⁺ ions. In Figure 5F and G, the authors also argue that the geometrical diameter and resulting curvature was the dominant factor limiting the kinetics of fusion pore opening. According to the plot of Fig. 5F, the fusion pore opening of the 40 nm-diameter synaptic vesicles has already reached its maximum speed and may not be further accelerated unless there is a change in the geometric parameters. The authors may want to comment on and discuss these issues that seem to be at odd with one another at a first glance.

3. If the loading of acridine orange dyes into vesicles is not new and used by many other authors, the main Figure 2 seems to be about technical details and the authors may want to move them to Supplementary Information.

Reviewer #3:

Remarks to the Author:

This manuscript by Kreutzberger et al. takes advantage of a reconstituted exocytosis system they previously developed to measure docking and fusion of three different types of native vesicles: neuronal synaptic vesicles, dense core vesicles containing neuropeptide-Y, and insulin-containing vesicles. They report critical components of each system necessary for docking or fusion.

Specifically, they observe changes in calcium dependence and the delay between calcium entry, docking, and fusion. Most significantly, they observe a correlation between the radius of curvature of the three types of vesicles and the delay time from calcium entry to fusion of docked vesicles.

The data match a previously proposed model. The findings are important and of general interest.

The revised manuscript clarifies several issues and provides additional data that strengthens the conclusions.

Response to Reviewers Revision 2

Reviewer #1 (Remarks to the Author):

In their revised manuscript, Kreutzberger and colleagues have satisfactorily addressed my previous concerns.

Minor point: In Figure 3B and Figure S4 please explain vertical orange and blue lines, respectively.

The vertical lines are the time determined for the arrival of calcium. This was added to the Figure legend. Line was recolored to orange in the Figure S4.

Reviewer #2 (Remarks to the Author):

The revised manuscript by Kreutzberger et al. addressed most of the concerns previously raised by the reviewers. In particular, the knock-in of single Syt isoforms in Fig. 5B and the titration of available v-SNAREs using soluble t-SNARE inhibitors (in Fig. 5C and D) demonstrate the power of this hybrid assay in dissection of the molecular mechanisms underlying the Ca²⁺-evoked exocytosis. This reviewer has only a few minor comments that the authors may find helpful in final preparation of their manuscript for publication in Nature Communications.

1. With the new experimental data using the soluble t-SNARE inhibitors, the authors suggest that the v-SNARE density may not critically determine the latency before the Ca²⁺-triggered fusion pore opening (Fig. 5C and D). On the other hand, this latency time dramatically depends on the presence of the C1C2MUN domain of Munc13 (Fig. 4). The current understanding in the field is that these MUNC13 domains work to facilitate initial assembly of the neuronal SNARE complexes. If the v-SNARE complex density (and probably, the number of neuronal SNARE complexes) is not a determining factor of the latency, this reviewer wonders what could be the molecular mechanism by which the MUNC13 domains shortens the latency. The authors may want to comment on and discuss this issue.

We added several sentences discussing this in blue to the second paragraph of the Discussion.

2. In Discussion, the authors argue that the fusion pore opening, in particular, of the synaptic vesicles was rate-limited by slow delivery of Ca²⁺ ions. In Figure 5F and G, the authors also argue that the geometrical diameter and resulting curvature was the dominant factor limiting the kinetics of fusion pore opening. According to the plot of Fig. 5F, the fusion pore opening of the 40 nm-diameter synaptic vesicles has already reached its maximum speed and may not be further accelerated unless there is a change in the geometric parameters. The authors may want to comment on and discuss these issues that seem to be at odd with one another at a first glance.

What we said in the paragraph is that the arrival of calcium requires a minimum time that is given by our injection method. Since calcium delivery was the same for all three types of

vesicles, the curvature-rate relationship still holds. We think that our text is sufficiently clear in this regard.

3. If the loading of acridine orange dyes into vesicles is not new and used by many other authors, the main Figure 2 seems to be about technical details and the authors may want to move them to Supplementary Information.

While the acridine orange loading itself has been done for many years, proof that the vesicle purification methods developed and used here have an active V-ATPase capable of generating the proton gradient and concentrating the dye is new and important for all subsequent figures of this manuscript. Therefore, we prefer to keep this figure as a main figure.

Reviewer #3 (Remarks to the Author):

This manuscript by Kreutzberger et al. takes advantage of a reconstituted exocytosis system they previously developed to measure docking and fusion of three different types of native vesicles: neuronal synaptic vesicles, dense core vesicles containing neuropeptide-Y, and insulin-containing vesicles. They report critical components of each system necessary for docking or fusion. Specifically, they observe changes in calcium dependence and the delay between calcium entry, docking, and fusion. Most significantly, they observe a correlation between the radius of curvature of the three types of vesicles and the delay time from calcium entry to fusion of docked vesicles. The data match a previously proposed model. The findings are important and of general interest. The revised manuscript clarifies several issues and provides additional data that strengthens the conclusions.

Prof. Dixon J. Woodbury
Brigham Young University

We thank the reviewer for his comments.